# Retiring △DP: New Distribution-Level Metrics for Demographic Parity

**Xiaotian Han**[1*], **Zhimeng Jiang**[1*], **Hongye Jin**[1*], **Zirui Liu**[2], **Na Zou**[1], **Qifan Wang**[3], **Xia Hu**[2]
*{han, zhimengj, jhy0410, nzou1}@tamu.edu, {zirui.liu, xia.hu}@rice.edu, wqfcr@meta.com*
[1] *Texas A&M University,* [2] *Rice University,* [3] *Meta AI*

*Reviewed on OpenReview:* *https://openreview.net/forum?id=LjDFIWWVVa*

## Abstract

Demographic parity is the most widely recognized measure of group fairness in machine learning, which ensures equal treatment of different demographic groups. Numerous works aim to achieve demographic parity by pursuing the commonly used metric $\Delta DP$ [1]. Unfortunately, in this paper, we reveal that the fairness metric $\Delta DP$ can not precisely measure the violation of demographic parity, because it inherently has the following drawbacks: *i)* zero-value $\Delta DP$ does not guarantee zero violation of demographic parity, *ii)* $\Delta DP$ values can vary with different classification thresholds. To this end, we propose two new fairness metrics, Area Between Probability density function Curves (ABPC) and Area Between Cumulative density function Curves (ABCC), to precisely measure the violation of demographic parity at the distribution level. The new fairness metrics directly measure the difference between the distributions of the prediction probability for different demographic groups. Thus our proposed new metrics enjoy: *i)* zero-value ABCC/ABPC guarantees zero violation of demographic parity; *ii)* ABCC/ABPC guarantees demographic parity while the classification thresholds are adjusted. We further re-evaluate the existing fair models with our proposed fairness metrics and observe different fairness behaviors of those models under the new metrics. The code is available at https://github.com/ahxt/new_metric_for_demographic_parity.

## 1 Introduction

Machine learning has been extensively adopted in various high-stake decision-making process, including criminal justice (Heidensohn, 1986; Berk et al., 2021; Tolan et al., 2019), healthcare (Ahmad et al., 2020; Cappelen & Norheim, 2006), college admission (Friedler et al., 2016), loan approval (Mukerjee et al., 2002; Kozodoi et al., 2022), and job marketing (Johnson et al., 2009; Raghavan et al., 2020). Since such high-stake decision-making could have a life-long effect on individuals, the fairness issue in these machine learning systems has raised increasing concerns (Chai et al., 2022; Zhang et al., 2022). Lots of fairness definitions (Garg et al., 2020; Mehrabi et al., 2021; Verma & Rubin, 2018; Saxena et al., 2019; Zafar et al., 2017; Mehrabi et al., 2020; Dwork et al., 2012; Hardt et al., 2016; Kleinberg et al., 2016) (e.g., demographic parity, equalized opportunity) has been proposed to solve different types of fairness issues. In this paper, we focus on the measurement of demographic parity, $\Delta DP$, which requires the predictions of a machine learning model should be independent on sensitive attributes (Menon & Williamson, 2018; Ustun et al., 2019; Kamishima et al., 2012).

To develop effective fair models, a faithful metric is critical to guide the development of a fair machine learning system since an "irrational" fairness metric may mislead the development of fair models and even lead to opposite conclusions. Despite the extensive efforts to develop various fairness metrics, one critical and fundamental question is still unclear: ***Are the existing metrics really reasonable to quantify demographic parity violation?***

---

[*]Equal contribution.
[1]$\Delta DP$ includes $\Delta DP_b$ and $\Delta DP_c$ in this paper, the details are presented in Section 2.1.

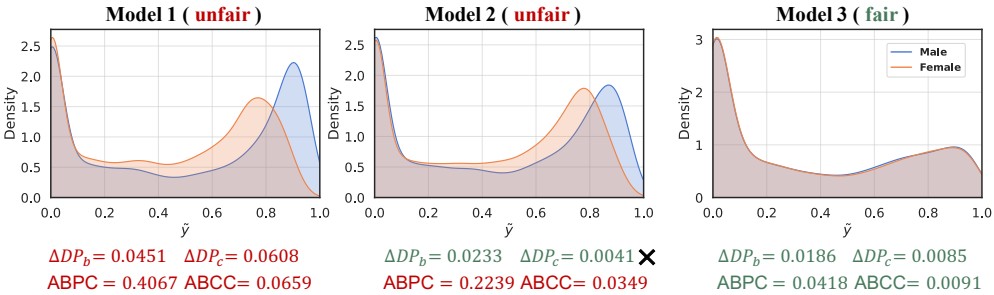

Figure 1: The distribution of the predictive probability of male and female groups on the ACS-Income dataset. $\Delta DP$ measures Model 1 and Model 3 correctly. However, it fails to assess (✗) Model 2 since it obtains a "fair" assessment on an "unfair" model. Our proposed metrics, ABPC and ABCC, can assess all the models precisely. Experimental results on more datasets are presented in Section 6.2.

In this paper, we rethink the rationale of $\Delta DP$ and investigate its limitations on measuring the violation of demographic parity. There are two commonly used implementations of $\Delta DP$ [2], including $\Delta DP_c$ (i.e., the difference of the mean of the predictive probabilities between different groups, used in papers (Chuang & Mroueh, 2020; Zemel et al., 2013)) and $\Delta DP_b$ (i.e., the difference of the proportion of positive prediction between different groups, used in papers (Dai & Wang, 2021; Creager et al., 2019; Edwards & Storkey, 2015; Kamishima et al., 2012)). We argue that $\Delta DP$, as a metric, has the following drawbacks: **First, zero-value $\Delta DP$ does not guarantee zero violation of demographic parity.** One fundamental requirement for the demographic parity metric is that the zero-value metric must be equivalent to the achievement of demographic parity, and vice versa. However, zero-value $\Delta DP$ does not indicate the establishment of demographic parity since $\Delta DP$ is a necessary but insufficient condition for demographic parity. An illustration of ACS-Income data is shown in Figure 1 to demonstrate that $\Delta DP$ fails to assess the violation of demographic parity since it reaches (nearly) zero on an unfair model (the middle subfigure in Figure 1). **Second, the value of $\Delta DP$ does not accurately quantify the violation of demographic parity and the level of fairness.** Different values of the same metric should represent different levels of unfairness, which is still true even in a monotonously transformed space. $\Delta DP$ does not satisfy this property, resulting in it being unable to compare the level of fairness based solely on its value. **Third, $\Delta DP_b$ value is highly correlated to the selection of the threshold for the classification task.** To make a decision based on predictive probability, one predefined threshold is needed. If the threshold for downstream tasks changes, the proportion of positive predictions of different groups will change accordingly, resulting in a change in $\Delta DP_b$ (Corbett-Davies et al., 2017; Menon & Williamson, 2017; Pleiss et al., 2017; Canetti et al., 2019). The selection of the threshold greatly affects the value of $\Delta DP_b$ (validated by Figures 2 and 7)(Chen & Wu, 2020; Barata et al.). However, threshold tuning is needed in practice. Adjusting the threshold is a common practice for decision-making but can violate demographic parity if the model is evaluated using $\Delta DP$ metric. There are several examples to illustrate dynamic threshold in the downstream task. For instance, in college admissions, the number of admissions and applicants can vary from year to year, necessitating adjustments to the decision-making threshold. Similarly, in AI-assisted medical diagnosis, doctors may modify the threshold for diagnosing a disease based on a patient's family history. However, if the model is evaluated using the $\Delta DP$ metric, demographic parity cannot be guaranteed if the threshold is changing. One specific $\Delta DP_b = 0$ can not guarantee demographic parity under the on-the-fly threshold change.

In view of the drawbacks of fairness metrics $\Delta DP$ for demographic parity, we first propose *two criteria* to theoretically guide the development of the metric on demographic parity: 1) Sufficiency: zero-value fairness metric must be a necessary and sufficient condition to achieve demographic parity. 2) Fidelity: The metric should accurately reflect the degree of unfairness, and the difference of such a metric indicates the fairness gap in terms of demographic parity. To bridge the gap between the criteria and the current demographic parity metric, we propose two distribution-level metrics, namely Area Between Probability density function Curves (ABPC) and Area Between Cumulative density function Curves (ABCC), to retire $\Delta DP_c$ and $\Delta DP_b$,

---

[2]We would like to claim that these two kinds of $\Delta DP$ are widely used to evaluate demographic parity in the current literature.

respectively. The advantage is that such independence can be guaranteed over any threshold, while $\Delta DP$ can only guarantee independence over a specific threshold.

The proposed metrics satisfy all (or partial) two criteria to guarantee the correctness of measuring demographic parity and address the limitations of the existing metrics, as well as estimation tractability from limited data samples. Moreover, we also re-evaluate the mainstream fair models with our proposed metrics. Our main contributions are as follows:

- We theoretically and experimentally reveal that the existing metric $\Delta DP$ for demographic parity can not precisely measure the violation of demographic parity, because it inherently has fundamental drawbacks: *i)* zero-value $\Delta DP$ does not guarantee zero bias, *ii)* $\Delta DP$ value does not accurately quantify the violation of demographic parity and *iii)* $\Delta DP$ value is different for varying thresholds.

- Motivated by the above observations, we formally established two criteria that a desirable metric on demographic parity should satisfy. This provides a guideline to assess other fairness metrics.

- We further propose two distribution-level metrics, Area Between Probability density function Curves (ABPC) and Area Between Cumulative density function Curves (ABCC), to resolve the limitations of $\Delta DP$, which are theoretically and empirically capable of measuring the violation of demographic parity.

- We re-evaluate the mainstream fair models with our proposed metrics, ABPC and ABCC, and re-assess their fairness performance on real-world datasets. Experimental results show that the inherent tension between fairness and accuracy is stronger, which has been underestimated previously.

It is worth noting that instead of invalidating the fairness definition of demographic parity, we claim that the current widely used metrics for demographic parity (i.e., $\Delta DP_b$ and $\Delta DP_c$) cannot precisely reflect the violation of demographic parity. In our paper, to precisely measure the violation of demographic parity, we propose two new and reasonable metrics, ABPC and ABCC, to measure the violation of demographic parity.

## 2 Preliminaries

**Notation.** Without loss of generality, we consider the binary classification task with binary sensitive attributes in this paper. We denote the dataset as $\mathcal{D} = \{(\mathbf{x}_i, y_i, s_i)\}_{i=1}^N$, where $N$ represents the number of samples, $\mathbf{x}_i \in \mathbb{R}^d$ is the features excluding sensitive attribute, $y_i \in \{0, 1\}$ is the label of the downstream task, and the $s_i \in \{0, 1\}$ is the sensitive attributes of $i$-th sample. The index set of groups with sensitive attributes is defined as $\mathcal{S}_0 = \{i : s_i = 0\}$ with $N_0$ samples, and $\mathcal{S}_1 = \{i : s_i = 1\}$ with $N_1$ samples. The predictive probability is denoted as $\tilde{y} \in [0, 1]$ by the machine learning model $f : \mathbb{R}^d \to [0, 1]$. The binary prediction is denoted as $\hat{y} \in \{0, 1\}$, where $\hat{y} = \mathbb{1}_{\geq t}[\tilde{y}]$, and $\mathbb{1}_{\geq t}(\cdot)$ represents the indicator function of whether is larger than threshold $t$. $X$ denotes the random variable that takes values $\mathbf{x}_i$. $Y$ denotes the random variable that takes values $y$. $S$ denotes the random variable that takes values $s$. $\hat{Y}$ denotes the random variable that takes values $\hat{y}$. We use $f_{0/1}(\cdot)$ and $F_{0/1}(\cdot)$ to denote the probability and cumulative distribution of the predictive value $\tilde{y}$ over the group with sensitive attribute $s = 0/1$, respectively.

### 2.1 Demographic Parity and Violation Measurement

The main idea behind demographic parity is that the prediction of the machine learning model should not be correlated to the sensitive attributes. Demographic parity can be achieved **if and only if the predictive probabilities are independent of the sensitive attributes**.

To measure the violation of demographic parity, several metrics has been proposed to evaluate the demographic parity. The widely used metrics are $\Delta DP_{continuous}$ (short for $\Delta DP_c$), which is calculated by the predictive probabilities (*continuous*), and $\Delta DP_{binary}^t$ (short for $\Delta DP_b^t$), which is calculated by the binary prediction (*binary*) with respect to a threshold $t$. Here we present their formal definitions.

$\Delta DP_b^t$ (Dai & Wang, 2021; Creager et al., 2019; Edwards & Storkey, 2015; Kamishima et al., 2012) is the difference between the proportion of the positive prediction between different groups, which uses the difference

of the average of the binary prediction between different groups as follows:

$$\Delta DP_b^t = \left| \frac{\sum_{n \in \mathcal{S}_0} \hat{y}_n^t}{N_0} - \frac{\sum_{n \in \mathcal{S}_1} \hat{y}_n^t}{N_1} \right|, \tag{1}$$

where the $\hat{y}_n^t \triangleq \mathbb{1}_{\geq t}[\tilde{y}_n]$ is the binary prediction of the downstream task based on pre-defined threshold $t$, $N$ is the number of the instances, and $N_{0/1}$ is the number of the samples in the group with sensitive attribute 0/1.

$\Delta DP_c$ (Chuang & Mroueh, 2020; Zemel et al., 2013) measures the difference of the average of the predictive probability between different demographic groups as follows:

$$\Delta DP_c = \left| \frac{\sum_{n \in \mathcal{S}_0} \tilde{y}_n}{N_0} - \frac{\sum_{n \in \mathcal{S}_1} \tilde{y}_n}{N_1} \right|, \tag{2}$$

where the $\tilde{y}$ is the prediction probability of the downstream task, $N$ is the total number of the instances, $N_{s=0/1}$ is the total number of the samples in the group with sensitive attribute 0/1. The key condition for $\Delta DP_c$ is that the average predictive probability $\tilde{y}$ among the same sensitive attribute group is a good approximation of the true conditional probability $P(\hat{Y} = 1|S = 0)$ or $P(\hat{Y} = 1|S = 1)$.

**Discussion** Relying solely on $\Delta DP_b^t$ and $\Delta DP_c$ as measurements for machine learning models can lead to trivial and unfair solutions, potentially misguiding the development of fair models. The current fairness methods usually result in trivial solutions if we use $\Delta DP_b^t$ and $\Delta DP_c$, as shown in the results of Model 2 in Figures 1 and 5. The "unfair" solution may obtain a lower $\Delta DP_b^t$ and $\Delta DP_c$, but with unfair prediction. However, $\Delta DP_b$ and $\Delta DP_c$ have become the *de facto* measurements for the current fairness metric, as many previous works use them as fairness metrics (e.g., [2][3] use $\Delta DP_b$ and [4][5] use $\Delta DP_c$). This could mislead the development of fairness methods in terms of demographic parity. $\Delta DP_b = 0$ (asymptotically) if and only if $\hat{y}$ is independent of the sensitive attributes since the conditional probability distribution $\hat{y}$ given sensitive attribute $S = 0$ and $S = 1$ are the same for the same ratio of positive samples across different demographic groups, i.e., $\Delta DP_c = \left| \frac{\sum_{n \in \mathcal{S}0} \hat{y}_n^t}{N_0} - \frac{\sum_{n \in \mathcal{S}1} \hat{y}_n^t}{N_1} \right| = 0$. However, we argue that our proposed metrics are a stronger version of $\Delta DP_b$, especially for dynamic thresholds. In other words, when our proposed metric ABCC or ABPC is zero, $\hat{y}$ is independent of the sensitive attributes for any threshold. Conversely, $\Delta DP_b^t = 0$ implies that $\hat{y}$ is independent of the sensitive attributes for specific thresholds. In a nutshell, our proposed fairness metrics are more general and stronger demographic parity metrics, which can be adopted in scenarios with dynamic thresholds.

**The relation between $\Delta DP_c$ and $\Delta DP_b^t$.** In the existing literature, $\Delta DP_c$ and $\Delta DP_b^t$ are both commonly used, where the former mainly focuses on the fairness over original predictive probability and the latter on fairness over the final decision making. Although these two metrics are adopted for different objectives, there is an intrinsic relation between them as shown in Theorem 2.1 (Please see Appendix A for more details.).

**Theorem 2.1** *The fairness measurement over original predictive probability $\Delta DP_c$ is upper bounded by the mean fairness measurement over binary prediction $\Delta DP_b^t$ with uniform-distribution threshold $t$, i.e., $\Delta DP_c \leq \int_0^1 \Delta DP_b^t dt$.. Additionally, there must exist a certain threshold $t^*$ so that the two fairness measurements are equivalent, i.e., $\Delta DP_c = \Delta DP_b^{t^*}$.*

## 3  Why does $\Delta$DP **Fail?**

In this section, we demonstrate that $\Delta DP$ is problematic in measuring the violation of demographic parity. We provide theoretical evidence of measurement failure of $\Delta DP$. In addition, we also empirically demonstrate that $\Delta DP$ cannot measure demographic parity properly.

### 3.1  Theoretical Analysis

**Argument 1: $\Delta DP = 0$ is only a necessary but insufficient condition for demographic parity to hold.** $\Delta DP$, relying on the difference between the average prediction or probability, is *not sufficiently*

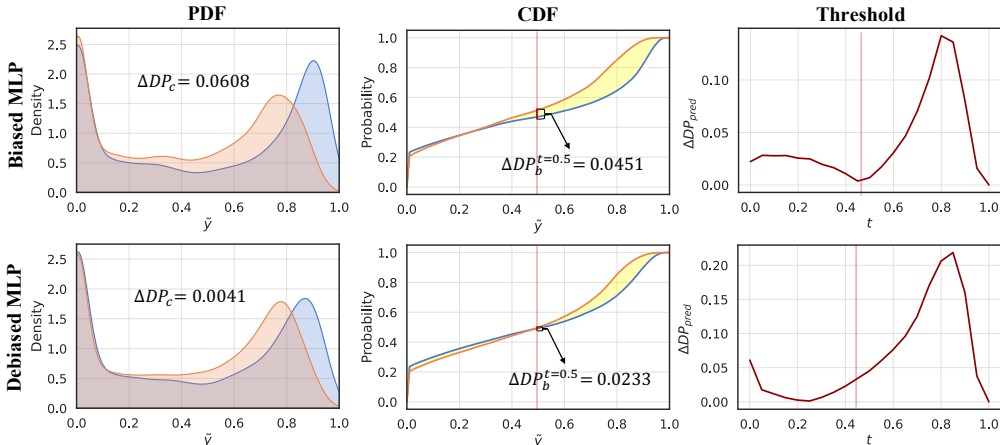

Figure 2: The estimated PDFs and empirical CDFs of the predictive probability over different groups (i.e., male, female) on the ACS-Employment dataset. *Top*: Biased MLP model. *Bottom*: Debiased MLP model with $\Delta DP_c$ as regularization. The PDFs and CDFs of biased MLP are obviously different, illustrating that the machine learning model is biased to gender. Debiased MLP achieves a much lower $\Delta DP_c$ than the biased MLP, however, the PDFs of different groups are different even though the "mean" of them are almost the same. The bottom figures show that $\Delta DP_b^t$ is the difference between the CDF lines while the threshold $t = 0.5$. We also provide the same set of results on Adult data in Appendix G.

*reliable* to quantify model prediction bias. According to characteristic function in probability theory (Sasvári, 2000), the same probability function is equivalent to the same $r$-th origin moment for any $r$. However, $\Delta DP$ only adopts average prediction (1-th origin moment) to define the metric. In other words, machine learning models can still be biased even if such metrics are zero. The insufficiency to guarantee demographic parity damages the authority of fairness measurement. In summary, $\Delta DP_c = 0$ and $\Delta DP_b^t = 0$ are necessary but insufficient conditions for demographic parity, which requires that the prediction should be independent of the sensitive attributes. We conclude that $\Delta DP_c = 0$ and $\Delta DP_b^t = 0$ are necessary but insufficient conditions for demographic parity, which requires that the prediction should be independent of the sensitive attributes [3]. We present a toy example to illustrate this argument. Let's consider we have the model's prediction $\hat{y} = [0.4, 0.4, 0.4, 0.4, 0.5, 0.5, 0.5, 0.5, 0.5, 0.9]$, and the sensitive attribute $s = [0, 0, 0, 0, 1, 1, 1, 1, 1, 0]$, We can see that the model is unfair since it tends to predict low values for samples with sensitive attribute 0, while predicting high values for samples with sensitive attribute 1, despite an outlier (the last sample). However, $\Delta DP_c = 0$, indicating a fair prediction. Thus, $\Delta DP_c$ may fail to measure fairness.

**Argument 2: Threshold Rules harm the measurement accuracy of $\Delta DP_b^t$.** In practice, the decision-making systems typically first predict a probability $\tilde{y}$ for the positive class and then obtain the binary prediction with a predefined threshold $t$, denoted as $\mathbb{1}_{\geq t}[\tilde{y}]$. For example, let $t = 0.5$, if $\tilde{y} \geq 0.5$ then the prediction is 1 else the prediction will be 0. This decision-making process is named *Threshold Rules* (Mitchell et al., 2021; Corbett-Davies & Goel, 2018). The establishment of $\Delta DP_b^t$ (Equation (1)) depends on the predefined threshold $t$ since the binary predictions $\hat{y}$ is determined by the threshold $t$. However, the so-called threshold rules could harm the measurement accuracy of $\Delta DP_b$. Changing the threshold for decision-making may lead to the changing demographic parity violation. The changing $\Delta DP_b^t$ highlights the fundamental drawbacks of the metric of demographic parity. We present a toy example to illustrate this argument. Let's consider the we have the model's prediction $\hat{y} = [0.35, 0.45, 0.55, 0.65]$ and the sensitive attribute $s = [0, 1, 0, 1]$, if we set the threshold to 0.5 for classification, $\Delta DP_b^{0.5} = 0$, indicating a fair prediction. However, if we set the threshold to 0.6, $\Delta DP_b^{0.6} = 0.67$, indication unfair prediction. Thus, with different thresholds, $\Delta DP_b^t$ may fail to measure fairness when the threshold changes.

## 3.2 Empirical Investigation

In this section, we empirically explore why $\Delta DP$ fails to measure the violation of demographic parity. We train one multilayer perceptron (Biased MLP) and the other MLP with fairness constraint (Debiased MLP)

---

[3]Please see more details in Appendix B.

on the UCI Adult and ACS-Employment dataset. Then we adopt Kernel Density Estimation (KDE) to estimate the PDFs of the predictive probability $\tilde{y}$ on the test dataset and also present the empirical CDFs. The results for ACS-Employment are presented in Figure 2, and the results on the Adult dataset are in Appendix G. From the experimental results, we have the following findings:

**Finding 1: The predictive probability distribution violates demographic parity.** On both the Adult and ACS-Employment datasets, the PDFs (Top-Left subfigure in Figure 2) are different among different groups, indicating the prediction is related to sensitive attributes. The difference between the PDFs reflects the degree of the violation of demographic parity. Similarly, the difference between the CDFs (Bottom-Left subfigure in Figure 2 ) also indicates the violation of demographic parity, and the area between the CDFs reflects the degree of fairness as well.

**Finding 2: $\Delta DP_b^t$ only measures the difference of the probability with a specific threshold $t$.** The CDFs (Middle-Left subfigure in Figure 2) vary among different groups, indicating the disparity of predictive probability distribution. Even with the debiased MLP, the CDFs (Middle-Right figures in Figure 2) still vary. Typically, the practitioners make decisions using 0.5 as a threshold for binary classification. Thus, the $\Delta DP_b^t$ is the difference of CDFs when $t = 0.5$, making the $\Delta DP_b^t$ an unreliable measurement for violation of demographic parity.

**Finding 3: $\Delta DP_b^t = 0$ with specific threshold $t$ can not guarantee the demographic parity.** On both the Adult and ACS-Employment datasets, the PDFs (Top-Right subfigure in Figure 2) are different among different groups but with the (nearly) same mean prediction, making $\Delta DP_b \approx 0$. This finding indicates that the $\Delta DP_b^t$ is a threshold-dependent metric, the value of which will be different if we choose different thresholds in downstream tasks.

## 4 Criteria for Fairness Measurement

> *Tell me how you measure me and I will tell you how I will behave. If you measure me in an illogical way. . . do not complain about illogical behavior. . .*
>
> — *Eliyahu Goldratt*

In this section, we provide a systematic understanding of fairness measurement from scratch, i.e., the criteria of fairness measurement design and intrinsic rationale. Given a dataset $\mathcal{D} = \{(\mathbf{x}_i, y_i, s_i)\}_{i=1}^n$ and well-trained model $\tilde{y} = f(\mathbf{x})$, the fairness measurement can be determined as $Bias(\mathcal{D}, f)$.

**Criterion 1 (Sufficiency): One metric should be necessary and sufficient to measure the demographic parity.** The fairness measurement is adopted to characterize the derivation of the model from the "ideal" unbiased one, therefore, the prediction value is independent of sensitive attributes *if and only if* the fairness metric is zero.

**Criterion 2 (Fidelity): One metric should be invariant with respect to monotone transformations of the distributions.** Different values of the fairness metric should represent different levels of fairness. As mentioned in the introduction, for example, if we change the scale on the predictive probability of one model, the fairness metric should be consistent in the original and scaled space. This property is referred to as *invariance over invertible transformation*. Thus we can compare the degree of fairness by comparing the values of $\Delta DP$. Inspired by this, we formally provide the following definition of invariance:

**Definition 4.1 (Invariance)** *For any invertible transformation $T : [0, 1] \to [0, 1]$, the fairness measurement $Bias(\mathcal{D}, f)$ satisfies measurement invariance condition if for any dataset $\mathcal{D}$ and machine learning model $f(\cdot)$, $Bias(\mathcal{D}, T \circ f) = Bias(\mathcal{D}, f)$ always holds.*

## 5 The Proposed Metrics

Following these criteria, in this section, we propose two metrics to measure the violation of demographic parity, namely Area Between PDF Curves (ABPC) and Area Between CDF Curves (ABCC), and prove that both ABPC and ABCC satisfy (or partially satisfy) the desired criteria.

## 5.1 ABPC and ABCC Metrics

In this section, we formally define two distribution-level metrics to measure the violation of demographic parity. ABPC is defined as the total variation distance ($TV$) between probability density functions with different sensitive attribute groups as follows:

$$\mathsf{ABPC} = TV(f_0(x), f_1(x)) = \int_0^1 |f_0(x) - f_1(x)|\, \mathrm{d}x, \tag{3}$$

where $f_0(x)$ and $f_1(x)$ are the PDFs of the predictive probability of different demographic groups. Similarly, ABCC is defined as the total variation between prediction cumulative density functions with different sensitive attribute groups as follows:

$$\mathsf{ABCC} = TV(F_0(x), F_1(x)) = \int_0^1 |F_0(x) - F_1(x)|\, \mathrm{d}x, \tag{4}$$

where $F_0(x)$ and $F_1(x)$ are the CDF of the predictive probability of demographic groups.

Note that the proposed metrics can be easily extended to the multi-value sensitive attribute setting. Suppose the sensitive attribute has $m$ values, then we compute the ABPC of each pair of groups with different sensitive attributes and then average them. ABCC for multi-value sensitive attributes with $m$ values can also be computed in a similar way. Since the $m$ is small in practice, the computational complexity is acceptable.

## 5.2 Theoretical Properties of ABPC and ABCC

In this section, we analyze the properties of ABPC and ABCC (Please see proofs in Appendix C), as well as their relation.

**Theorem 5.1 (Properties of ABPC)** *The proposed ABPC has the following desired properties: 1) ABPC = 0 holds if and only if demographic parity is established. 2) ABPC is invariant to invertible transformation. 3) ABPC has a range of $[0, 2]$.*

**Relation to $\Delta\mathrm{DP}_c$:** The proposed fairness metric ABPC upper bounds $\Delta\mathrm{DP}_c$, i.e., $\mathsf{ABPC} \geq \Delta\mathrm{DP}_c$, which overcomes the drawback of $\Delta\mathrm{DP}_c$ regarding the insufficient condition of demographic parity. Please see Appendix D for more details.

**Theorem 5.2 (Properties of ABCC)** *The proposed ABCC has the following desired properties: 1) ABCC = 0 if and only if demographic parity establishes. 2)ABCC is continuous over model prediction. 3) ABCC has a range $[0, 1]$.*

**Relation to $\Delta\mathrm{DP}_b^t$:** The ABCC generalizes $\Delta DP_b^t$. $\Delta DP_b^t$ is the difference of positive prediction proportion over different groups with respect to threshold rule $t$, thus $\Delta DP_b^t = |F_0(t) - F_1(t)|$. Considering the definition of $\mathsf{ABCC} = \int_0^1 |F_0(x) - F_1(x)|\, \mathrm{d}x$, we have $\mathsf{ABCC} = \int_0^1 \Delta DP_b^t dt$. This indicates that ABCC is the average of the $\Delta DP_b^t$ over all possible threshold $t$.

**Relation between ABPC and ABCC.** Hereby we show the relation between ABPC and ABCC that they are both Wasserstein distance between the prediction PDFs of two groups, but with different transport cost functions (See Appendix E for more details). The proposed fairness metric ABPC is Wasserstein distance with cost function $c_0(x, y) = 2 \cdot \mathbb{1}(x \neq y)$, defined as $W_{c_0}(f_0(x), f_1(y))$. The proposed fairness metric ABCC is Wasserstein distance with $l_1$ cost $c(x, y) = |x - y|$ between the prediction PDFs with different sensitive attributes. In other words, ABPC and ABCC can be interpreted as Wasserstein distance between the PDFs for different sensitive attribute groups with different transport cost functions.

## 5.3 Estimating ABPC and ABCC

In real-world scenarios, the fairness measurement is based on the PDFs and CDFs of predictive probability, which can be estimated from finite data samples. Hereby we present the estimation methods for both ABPC and ABCC and present the theoretical and empirical analysis for the estimation.

*For the proposed* ABPC, the PDFs can be estimated via kernel density estimation (KDE) given predictive probability $\{\tilde{y}_n, n \in \mathcal{S}_i\}$ for each sensitive attribute group with $s = i$. The basic idea for PDF estimation is that the prediction value for each sample represents a local PDF component (kernel function) and the overall mixed local PDF is the estimated PDF. Given the smoothing kernel function $K(x)$ (e.g., Gaussian kernel) satisfying normalization condition $\int K(x)\mathrm{d}x = 1$, and bandwidth $h$, then the estimated PDF is $\tilde{f}_i(x) = \frac{1}{|\mathcal{S}_i|h} \sum_{n \in \mathcal{S}_i} K(\frac{x - \tilde{y}_n}{h})$, for $i = 1, 2$. Subsequently, ABPC can be estimated via Equation (3).

*For the proposed* ABCC, we directly adopt the empirical distribution function (Shorack & Wellner, 2009) as estimated CDF, which measures the fraction of samples' predictions that are less or equal to the specified threshold. Formally, given predictive probability $\{\tilde{y}_n, n \in \mathcal{S}_i\}$, the empirical distribution function $\hat{F}_i$ for sensitive attribute $s = i$ is given by $\hat{F}_i(x) = \frac{1}{N_i} \sum_{n \in \mathcal{S}_i} \mathbb{1}_{\leq x}(\tilde{y}_n)$, where $i = 0, 1$. Hence, based on the definition of proposed metrics, we have $\hat{\mathsf{ABPC}} = \int_0^1 |\hat{f}_0(x) - \hat{f}_1(x)|\mathrm{d}x$ and $\hat{\mathsf{ABCC}} = \int_0^1 |\hat{F}_0(x) - \hat{F}_1(x)|\mathrm{d}x$. Firstly, we provide the following definition to measure estimation tractability with finite data samples as follows:

**Definition 5.3 ($(N, \epsilon)$-tractability)** *Given the dataset $\mathcal{D}$ with $N$ data samples, underlying yet unknown data distribution $P_\mathcal{D}$, and well-trained machine learning model $f(\cdot)$, the fairness measurement satisfies $(N, \epsilon)$-tractability if for any dataset $\mathcal{D}$ and machine learning model $f(\cdot)$, the mean square estimation error condition $\mathbb{E}_\mathcal{D}[|Bias(\mathcal{D}, f) - Bias(P_\mathcal{D}, f)|^2] \leq \epsilon$ holds.*

Subsequently, we provide theoretical analysis on the estimation error for estimated metrics, $\hat{\mathsf{ABPC}}$ and $\hat{\mathsf{ABCC}}$, as follows (more details in Appendix F):

**Lemma 5.4 (Estimation Tractability)** *Suppose we adopt KDE to estimate the prediction probability density function and directly calculate the estimated fairness metrics $\hat{\mathsf{ABPC}}$ and $\hat{\mathsf{ABCC}}$, such fairness metrics estimation satisfies $(N, O(N^{-\frac{4}{5}}))$-tractability and $(N, O(N^{-1}))$-tractability, where $N$ represents the number of data samples. Furthermore, for the number of samples $N = O(\delta^{-\frac{5}{4}}\epsilon^{-\frac{5}{2}})$ and $N = O(\delta^{-1}\epsilon^{-2})$ for $\hat{\mathsf{ABPC}}$ and $\hat{\mathsf{ABCC}}$, with probability at least $1 - \delta$, $|Bias(\mathcal{D}, f) - Bias(\mathcal{P}_\mathcal{D}, f)| < \epsilon$.*

Lemma 5.4 demonstrates the reliability of our proposed metric with the finite data samples using the proposed estimation method. Recall demographic parity requires independent prediction with respect to sensitive attributes, such independence should be guaranteed at the distribution level. However, we only observe limited data samples, instead of the prediction distribution. Hence, the fairness measurement must be reliably estimated from limited observed data samples. The estimation error convergence provides a dispensable foundation for fairness measurement and model comparison in terms of fairness in practice.

**Computation Complexity.** The computation of the proposed ABPC and ABCC metrics, defined in Equations (3) and (4), are based on estimated probability density function $\tilde{f}_i(x)$ and cumulative function $\tilde{F}_i(x)$, and numerical integration over $[0, 1]$ with $M$ probing points. For ABPC, the estimation of PDFs is given by $\tilde{f}_i(x) = \frac{1}{|\mathcal{S}_i|h} \sum_{n \in \mathcal{S}_i} K(\frac{x - \tilde{y}_n}{h})$. The computation complexity of $\tilde{f}_i(x)$ with $M$ probing points is $O(MN)$, where $N$ is the number of samples, and the computation complexity of numerical integration is $O(M)$. Therefore, the computation complexity of ABPC is $O(MN)$. For ABCC, the computation complexity for the estimation of CDF with $M$ probing points and numerical integration is $O(MN)$ and $O(M)$, respectively. Thus, the computation complexity of ABCC is also $O(MN)$.

**Discussion about Direct Optimization of ABPC and ABCC.** The proposed ABPC and ABCC metrics are both differentiable w.r.t. model parameters and thus can be directly optimized. The key reason is that the (conditional) prediction probability density can be estimated based on KDE, i.e., $\tilde{f}_i(x) = \frac{1}{|\mathcal{S}_i|h} \sum_{n \in \mathcal{S}_i} K(\frac{x - \tilde{y}_n}{h})$, where $K(x)$ is smoothing Gaussian kernel function, $\tilde{y}_n$ is the model prediction for $n$-th sample, and $h$ is pre-defined bandwidth. Note that ABPC is differentiable w.r.t. (conditional) prediction probability density, and (conditional) prediction probability density is differentiable w.r.t. model prediction and model parameters, ABPC can be directly optimized. Similarly, ABCC also can be directly optimized. We also clarify that this paper mainly focuses on the fairness metric side, the bias mitigation method development for ABPC and ABCC can be left for future work.

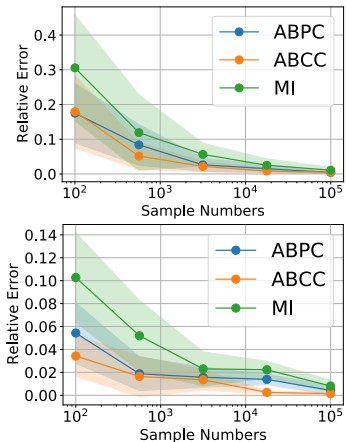 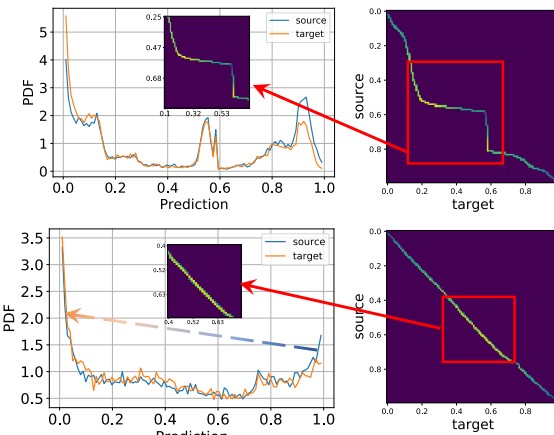

Figure 3: Relative estimation error for ABPC, ABCC, and mutual information. *Top:* The synthetic data is from $\mathcal{N}(0.3, 0.1)$ and $\mathcal{N}(0.4, 0.1)$. *Bottom:* The synthetic data is from $\mathcal{N}(0.2, 0.1)$ and $\mathcal{N}(0.4, 0.1)$.

Figure 4: Bias density visualization. *Top:* The source and target distributions and bias density visualization for vanilla MLP model. *Bottom:* The source and target distributions and bias density visualization for adversarial debiasing. The fair model represents the diagonal optimal transport plan, i.e., the source and target distribution is aligned everywhere.

## 5.4 Comparison with Related Work

We comprehensively analyze the existing demographic parity metrics and provide their inherent relations and differences. In other words, our work is metric-centric and provides insights to identify the promising properties, namely sufficiency, and fidelity, of fairness metrics. We highlight the differences with works (Jiang et al., 2020; 2022b) in terms of fairness metric and estimation tractability.

- **Difference with Jiang et al. (2020)**: For fairness metric, the motivation and justification differ significantly. In this work, we start with the proposed two criteria (i.e., sufficiency, fidelity), and then design fairness metrics as the distance of probability density function (PDF) and cumulative distribution function (CDF). Furthermore, we analyze that the proposed fairness metrics are actually some version of Wasserstein distance. Instead, (Jiang et al., 2020) directly starts with Wasserstein distance to enforce strong demographic parity. For metric justifications, we mainly focus on the analysis of whether the proposed two criteria hold or not. Instead, (Jiang et al., 2020) provides an in-depth analysis of Wasserstein distance and demonstrates several equivalent formats in Section 3.1. On top of the analysis, Wasserstein penalized logistic regression method and post-processing method are proposed to achieve fair prediction.

- **Difference with Jiang et al. (2022b)**: The results on estimation tractability is similar to (Jiang et al., 2022b). However, we clarify that the target metrics are differ significantly. Jiang et al. (2022b) propose GDP to measure the bias for continuous sensitive attributes via the difference of average prediction across different sensitive attributes. Instead, we only focus on the proposed two fairness metrics for binary sensitive attributes by measuring the distribution distance. Our result shows that the proposed metrics have the same estimation tractability as that in GDP. The main reason is that the estimation method of our paper and Jiang et al. (2022b) are both derived from non-parametric estimation theory (Sampson & Guttorp, 1992). These results demonstrate the proposed metrics can be reliable estimations in practice.

## 5.5 Experimental Evaluation

We empirically evaluate the estimation tractability of our proposed fairness metrics and visualize the bias density. First, we show the relative estimation error of our proposed metrics is lower than that of mutual information (MI) in the experiments with synthetic data. Subsequently, we visualize the bias density for vanilla MLP and adversarial debiasing method in ACS-Income dataset.

### 5.5.1 Estimation Tractability on Synthetic Data

We test the relative estimation error of the proposed ABPC and ABCC compared with mutual information, which precisely measures the dependence between two random variables. For data generation, we first generate Gaussian mixture distribution with different means and variances for different groups. Then we use the sigmoid function to map all data into $[0, 1]$. For metric estimation, we adopt KDE and empirical CDF to estimate the ground-truth PDF and CDF, and then directly calculate estimated metrics via numerical integration. The relative estimation error is defined as the deviation of the estimated metric by ground-truth metrics.

Figure 3 shows the relative estimation error of different fairness metrics with respect to sample numbers for the synthetic data with different parameters. We observe that the relative estimation error for ABCC and mutual information are the lowest and the largest for different numbers of data samples, respectively. In other words, our proposed metrics embrace higher estimation tractability with finite data samples.

In this experiment, mutual information is calculated based on prediction probability and conditional prediction probability, i.e., $MI(\tilde{Y}; S) = H(\tilde{Y}) - H(\tilde{Y}|S) = H(\tilde{Y}) - \mathbb{P}(S = 0)H(\tilde{Y}|S = 0) - \mathbb{P}(S = 1)H(\tilde{Y}|S = 1)$, where the entropy function $H(\tilde{Y}) = \int_0^1 f_{\tilde{Y}}(\tilde{y})\mathrm{d}\tilde{y}$, and $H(\tilde{Y}|S = i) = \int_0^1 f_{\tilde{Y}|S=i}(\tilde{y})\mathrm{d}\tilde{y}$ for $i = 0, 1$. Mutual information between model prediction $\tilde{Y} \in [0, 1]$ and sensitive attribute $S \in 0, 1$ can be adopted to measure the independence. Compared with the definitions of ABPC and ABCC, the calculation of MI is also based on the estimated (conditional) prediction probability, as shown in Section 5.3. The advantage of our proposed ABPC and ABCC is the lower metric relative estimation error.

### 5.5.2 Bias Density Visualization

Section 5.2 shows that ABCC is the $1^{st}$-Wasserstein distance of two PDFs of different groups (Please see Appendix I for more details). In other words, ABCC can be decomposed into the integration of bias density over prediction domains via calculating the optimal transport plan. Suppose $\gamma^*(x, y)$ is the optimal transportation plan between these two PDFs. For ABCC, we can define the *bias density* as $\rho(x, y) \triangleq |x - y|\gamma^*(x, y)$ since ABCC satisfies ABCC $= \int \rho(x, y)\mathrm{d}x\mathrm{d}y$. In other words, the bias can be decomposed across the prediction value domain for these two sensitive attribute groups. We visualize the bias density of the vanilla method and adversarial debiasing on ACS-Income dataset. Figure 4 shows the estimated PDFs for two groups (source and target distribution) and visualizes the bias density. For the top subfigure, the source distribution is higher than the target distribution at around 0.5 and 0.9 prediction values. The bias density shows that this excess part at the source distribution should be moved toward the target distribution at around 0.2 and 0.6 prediction values. For the bottom subfigure, these two distributions are extremely close and thus lead to a low fairness metric. The optimal transport plan is close to the diagonal line, demonstrating that most parts of the source distribution keep the original value.

## 6 Re-evaluating Existing Fair Models

In this section, we conduct experiments on various datasets to re-evaluate the commonly-used fair models.

### 6.1 Experimental Setting

**Debiasing Methods.** We consider the vanilla MLP model (MLP) and widely used debiasing methods, including regularization (REG), and adversarial debiasing (ADV). **MLP** (Multilayer Perceptron) is the vanilla multilayer perceptron to minimize the empirical risk of downstream tasks. Thus MLP tends to be biased the underprivileged group, making it a biased model. In our experiments, we adopt a 4-layer fully-connected network and utilize ReLU (Nair & Hinton, 2010) as the activation function. The same network architecture is also adopted as the classification network for other baselines. **ADV** (Adversarial Debiasing) (Louppe et al., 2017) jointly trains a classification network and an adversarial network. The adversarial network takes the output of the classifier as its input and is trained to distinguish which sensitive attribute group the output comes from. We train the classifier to provide correct predictions for the input data while training the adversarial network not to distinguish groups from the output of this classifier simultaneously.

Table 1: The fairness performance on the tabular dataset for existing fair models and we consider race and gender as sensitive attributes. ↑ represents the accuracy improvement compared to MLP. A higher accuracy metric indicates better performance. ↓ represents the improvement of fairness compared to MLP. A lower fairness metric indicates better fairness. The results are based on 10 runs for all methods.

| | Methods | | Accuracy | | | | Fairness | | | | | | | |
|---|---|---|---|---|---|---|---|---|---|---|---|---|---|---|
| | | | Acc(%) | ↑ | AP(%) | ↑ | $\Delta DP_b^t$(%) | ↓ | $\Delta DP_c$(%) | ↓ | ABPC(%) | ↓ | ABCC(%) | ↓ |
| **Adult** | Race | MLP | 85.54±0.19 | — | 77.33±0.27 | — | 19.27±0.59 | — | 20.03±0.44 | — | 64.65±1.07 | — | 20.03±0.44 | — |
| | | REG | 85.31±0.12 | -0.27% | 75.55±0.15 | -2.30% | 14.18±0.67 | 26.41% | 0.81±0.52 | 95.96% | 62.21±2.53 | 3.77% | 10.79±0.53 | 46.13% |
| | | ADV | 80.03±1.78 | -6.44% | 61.82±4.84 | -20.06% | **3.89**±1.94 | 79.81% | **1.62**±1.41 | 91.91% | **21.26**±3.75 | 67.12% | **2.60**±1.06 | 87.02% |
| | Gender | MLP | 85.52±0.12 | — | 77.33±0.27 | — | 21.84±0.59 | — | 25.93±0.60 | — | 98.62±0.93 | — | 25.93±0.60 | — |
| | | REG | 85.03±0.22 | -0.57% | 73.55±0.65 | -4.89% | 15.58±0.69 | 28.66% | 1.30±1.05 | 94.99% | **80.62**±4.62 | 18.25% | 12.86±0.30 | 50.40% |
| | | ADV | 76.34±0.61 | -10.73% | 69.38±2.89 | -10.28% | **0.31**±0.69 | 98.58% | **0.56**±0.72 | 97.84% | 87.92±43.22 | 10.85% | **0.56**±0.72 | 97.84% |
| **KDD Census** | Race | MLP | 94.94±0.04 | — | 99.50±0.00 | — | 2.64±0.11 | — | 3.73±0.08 | — | 18.91±0.58 | — | 3.73±0.08 | — |
| | | REG | 94.81±0.05 | -0.14% | 99.42±0.01 | -0.08% | 1.61±0.21 | 39.02% | 0.78±0.27 | 79.09% | **10.19**±1.43 | 46.11% | 0.83±0.26 | 77.75% |
| | | ADV | 93.72±0.29 | -1.29% | 99.13±0.16 | -0.37% | **0.14**±0.18 | 94.70% | 0.32±0.28 | 91.42% | 11.11±9.25 | 41.25% | **0.34**±0.27 | 90.88% |
| | Gender | MLP | 94.84±0.05 | — | 99.45±0.00 | — | 4.75±0.36 | — | 5.99±0.30 | — | 40.96±0.99 | — | 5.99±0.30 | — |
| | | REG | 94.45±0.06 | -0.41% | 99.31±0.03 | -0.14% | 1.38±0.20 | 70.95% | 0.86±0.23 | 85.64% | 8.57±0.19 | 79.08% | 1.04±0.14 | 82.64% |
| | | ADV | 93.66±0.17 | -1.24% | 98.62±0.24 | -0.83% | **0.35**±0.27 | 92.63% | **0.36**±0.26 | 93.99% | **5.11**±2.69 | 87.52% | **0.45**±0.22 | 92.49% |
| **ACS-I** | . Race | MLP | 81.80±0.10 | — | 84.83±0.15 | — | 9.57±0.24 | — | 7.47±0.12 | — | 16.82±0.34 | — | 7.47±0.12 | — |
| | | REG | 81.15±0.18 | -0.79% | 83.92±0.12 | -1.07% | 3.11±0.43 | 67.50% | 1.66±0.37 | 77.78% | 9.19±0.52 | 45.36% | 2.48±0.18 | 66.80% |
| | | ADV | 77.72±0.28 | -4.99% | 79.06±0.39 | -6.80% | **0.45**±0.50 | 95.30% | **0.26**±0.26 | 96.52% | **2.78**±0.74 | 83.47% | **0.38**±0.19 | 94.91% |
| | Gender | MLP | 81.78±0.09 | — | 84.65±0.16 | — | 9.14±0.16 | — | 7.97±0.11 | — | 14.74±0.59 | — | 7.97±0.11 | — |
| | | REG | 80.93±0.05 | -1.04% | 83.61±0.17 | -1.23% | 2.10±0.22 | 77.02% | 1.60±0.20 | 79.92% | 3.69±0.42 | 74.97% | 1.60±0.20 | 79.92% |
| | | ADV | 78.42±0.85 | -4.11% | 79.62±0.42 | -5.94% | **0.32**±0.20 | 96.50% | **0.24**±0.14 | 96.99% | **2.54**±0.72 | 82.77% | **0.48**±0.10 | 93.98% |
| **ACS-E** | Gender | MLP | 81.78±0.04 | — | 85.26±0.05 | — | 5.49±0.79 | — | 6.73±0.47 | — | 42.18±0.70 | — | 7.12±0.44 | — |
| | | REG | 81.57±0.06 | -0.26% | 84.51±0.13 | -0.88% | **1.02**±0.65 | 81.42% | **0.39**±0.33 | 94.21% | 22.40±1.06 | 46.89% | 3.30±0.12 | 53.65% |
| | | ADV | 77.71±3.59 | -4.98% | 81.39±0.36 | -4.54% | 1.15±0.74 | 79.05% | 0.82±0.77 | 87.82% | **3.67**±0.15 | 91.30% | **0.97**±0.62 | 86.38% |

**REG** (Regularization) is a kind of in-process method that adds a fairness-related regularization term to the objective function (Chuang & Mroueh, 2020; Kamishima et al., 2012). This kind of method improves the fairness of the model with the regularization term simultaneously optimized during training. In our experiments, REG takes $\Delta DP_c$ as the regularization term. The objective function is defined as $\mathcal{L}_{ce} + \lambda\mathcal{L}_{dp}$, where $\mathcal{L}_{ce}$ is the cross-entropy loss for downstream task and $\mathcal{L}_{dp}$ is fairness constraint (Equation (2)).

**Dataset.** We consider the following datasets in our experiment. including tabular dataset and image dataset (See more experimental results in Appendix H). **UCI Adult** (Dua & Graff, 2017) contains clean information about $45,222$ individuals from the 1994 US Census. One instance is described with 15 attributes. The task is to predict whether the income of a person is higher than $50k given attributes about the person. We considered gender and race as sensitive attributes. **ACS-I**ncome (Ding et al., 2021) derives from the American Community Survey (ACS) Public Use Microdata Sample (PUMS). Like UCI Adult, the task on this dataset is to predict whether an individual's income is above $50k. The dataset contains $1,664,500$ data points. We choose gender and race as sensitive attributes. **ACS-E**mployment (Ding et al., 2021) also derives from the ACS PUMS. This task is to predict whether an individual is employed. The number of data in this dataset is $3,236,107$. In this dataset, gender is used as the sensitive attribute. **KDD Census** (Dua & Graff, 2017) contains $284,556$ clean instances with 41 attributes. The task of this dataset is also to predict whether the individual's income is above $50k. The sensitive attributes are gender and race.

**Evaluation Metric.** The fairness metrics are $\Delta DP_c$, $\Delta DP_b$, ABPC and ABCC. The evaluation metrics for model accuracy are: 1) **Acc**: the accuracy is the fraction of correct predictions; 2) **AP**: the average precision (AP) is defined as $AP = \sum_n (R_n - R_{n-1})P_n$ where $P_n$ and $R_{n-1}$ are the precision and recall at the $n$-th threshold. The value is between 0 and 1, and higher is better.

## 6.2 Will ABPC and ABCC Measure the Violation of Demographic Precisely?

We empirically validate the effectiveness of the proposed metrics in this section. We train three different models and plot the distribution of the predictive probability for different groups, and we report the fairness

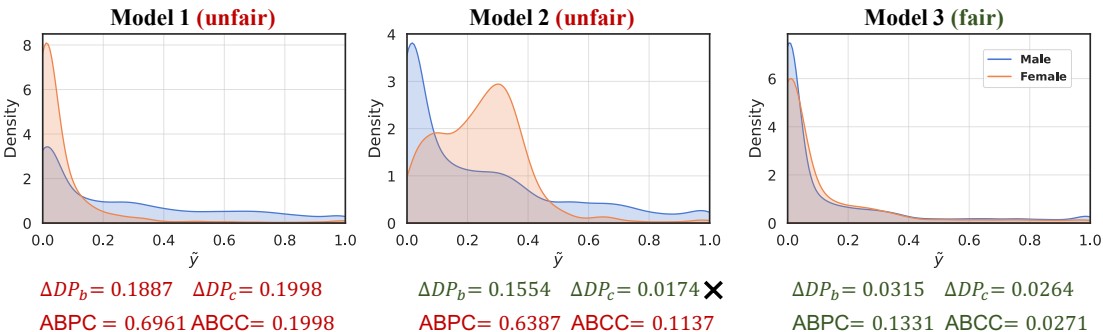

Figure 5: The distribution of the predictive probability of male and female groups on the Adult dataset.

metrics $\Delta DP_b^t$ and $\Delta DP_c$, ABPC and ABCC for them in Figures 1 and 5. Figures 1 and 5 show the predictive probability distribution of three machine models. The Models 1, 2, and 3 are unfair, unfair, and fair, respectively, since the distributions of Model 2 for different groups are the same while others are not. One can see that $\Delta DP$ can measure the unfair model (Model 1) and fair model (Model 3) correctly since it has a larger $\Delta DP$ value for Model 1 than Model 3. However, it fails to assess Model 2 (unfair) since it obtains a relatively low value on an unfair model. Our proposed metrics ABPC and ABCC can assess all the models correctly. The wrong fairness assessment on Model 2 demonstrates the $\Delta DP$ is problematic in measuring the violation of demographic parity. $\Delta DP_b$ was able to measure the violation of Model 2 in Figure 5 on the Adult dataset, where $\Delta DP_b = 0.1554$ indicates that the model is unfair. However, for Model 2 in Figure 1 on the ACS-Income dataset, both $\Delta DP_b$ and $\Delta DP_c$ failed to evaluate the demographic parity violation. Therefore, $\Delta DP_b$ cannot consistently evaluate models across different datasets, which is another experimental evidence of its failure to measure the violation of demographic parity. Regarding the model side, this may be due to the fact that current fairness methods often result in trivial solutions when using $\Delta DP_b$ and $\Delta DP_c$ as measurements.

### 6.3   How the Existing Fair Models Performs with ABPC and ABCC?

We conduct experiments on the aforementioned four datasets and baselines. Since there is no universal best model that optimizes both fairness and accuracy objectives, we trained all the models with a fixed number of epochs and reported the performance on the test dataset. We train MLP and REG for 10 epochs and train ADV for 40 epochs. We use ten different dataset splits and report the mean and standard deviation of Acc and AP. We report the prediction performance and the fairness performance of the downstream task in Table 1. We have the following **Obs**ervations:

**Obs. 1: REG method always achieves a lower $\Delta DP_c$ but a relatively high $\Delta DP_b^t$.** Since the REG directly optimizes the $\Delta DP_c$, REG always obtains a lower $\Delta DP_c$. However, REG achieves much higher ABPC and ABCC than other methods.

**Obs. 2: ADV is a better fair model to achieve demographic parity.** In terms of the proposed metrics, the ADV gains 6 better ABPC among 8 cases and 8 better ABCC among 8 cases, showing that ADV achieves the best fairness performance. The reason why ADV performs better using our proposed metrics is twofold. First, adversarial learning can intuitively and theoretically ensure that the predicted probabilities (a continuous value) are independent of the sensitive attributes. This is supported by both theoretical and empirical evidence. Second, our two proposed metrics are designed to be 0 only when the predicted probabilities are fully independent of the sensitive attributes. This provides a more strict and precise measure of demographic parity, which may make it easier for the ADV method to achieve better performance on this metric. On the other hand, the REG method in our paper uses $\Delta DP_c$ as a fairness regularization term (as well as a surrogate loss function for $\Delta DP_c$). By using the difference of the average predicted probabilities for demographic groups, it can ensure that the predicted probabilities are fully independent of the sensitive attributes. Thus, it may achieve secondary performance on our proposed metric. The experimental observation that ADV performs better with respect to their metrics somehow shows that

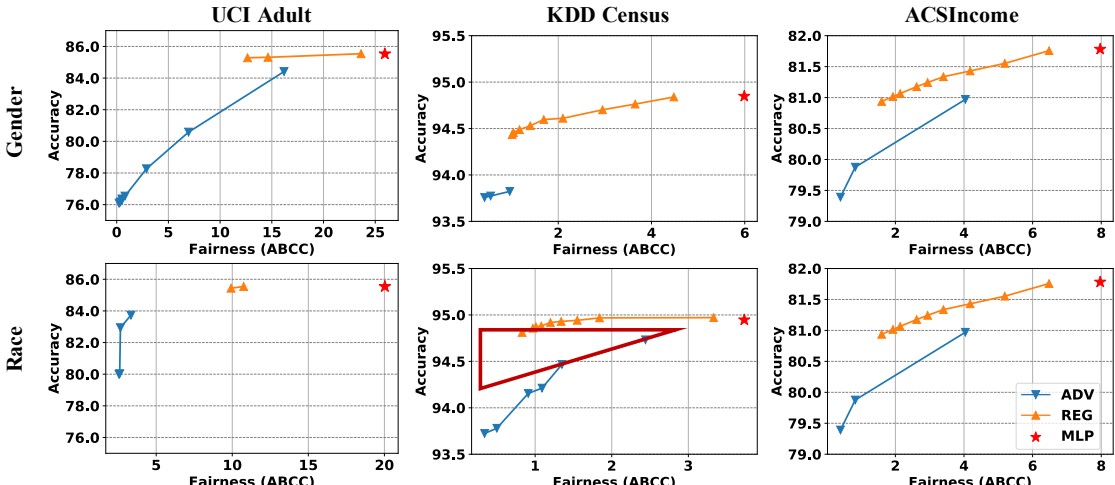

Figure 6: The accuracy and fairness trade-off on the tabular dataset. The Pareto frontiers show the accuracy-fairness trade-off of different fair models.

measuring demographic parity from the distribution of predicted probabilities can provide more insight into the behavior of existing fairness methods.

**Obs. 3: The inherent tension between fairness and accuracy is underestimated.** It is the prevailing wisdom that a model's fairness and its accuracy are in tension with one another. Although the REG does not harm the model accuracy dramatically, it can not achieve ideal fairness. In contrast, the ADV achieves better fairness but harms the model accuracy dramatically. This observation demonstrates that if we tend to achieve fair decision-making, we have to sacrifice more accuracy.

### 6.4 How Accuracy and Fairness Trade-off with the Proposed Metrics?

In this section, we provide the "Pareto front" to investigate the tension between the model accuracy (Acc) and fairness (ABCC) shown in Figure 6. For REG, we set different values of the trade-off hyperparameter $\lambda \in [0, 1]$ to control the accuracy-fairness trade-off and $\lambda \in [10, 180]$ for ADV. We have the following **Obs**ervations:

**Obs. 1: REG tends to gain a better accuracy performance while ADV tends to gain a better fairness performance.** The Pareto front of REG is higher than that of ADV, indicating that REG obtains a better accuracy performance. In contrast, the Pareto front of ADV is lower than that of REG, but it can reach a lower ABCC area, indicating that REG obtains a better fairness performance. The results show that our proposed metrics can better evaluate the performance of fair models and provide new standards to analyze model debiasing.

**Obs. 2: The inherent tension between fairness and accuracy is underestimated.** The REG method cannot achieve a lower ABCC, which may limit its application to real-world tasks. Although the ADV can achieve a much lower ABCC, the accuracy drop is too high. In other words, ADV sacrifices too much performance in pursuit of fairness. The area (indicated as ▷) points out the potential direction of the fair model development.

## 7 Related Work

In this section, we present related works about fairness and metrics for demographic parity.

**Fairness.** Algorithmic fairness is legally mandatory in various high-stake real-world applications. The various fairness definitions have been proposed and categorised into *individual fairness* (Yurochkin et al., 2019; Mukherjee et al., 2020; Yurochkin & Sun, 2020; Kang et al., 2020; Mukherjee et al., 2022), *group fairness* (Hardt et al., 2016; Verma & Rubin, 2018; Li et al., 2020; Ling et al., 2023; Jiang et al., 2022a), and *counterfactual fairness* (Kusner et al., 2017; Agarwal et al., 2021; Zuo et al., 2022). In this paper, we focus

on demographic parity, a widely studied group fairness. The metric for group fairness measures the disparity between subgroups defined by sensitive attributes (e.g., gender and race). For example, demographic parity (Dwork et al., 2012) aims to render the independence of the prediction and sensitive attribute, and $\Delta DP$ measuring the average prediction disparity among different groups is widely adopted as the fairness metrics for demographic metrics. Several works leverage other measurements (i.e., Wasserstein distance, AUC) for bias mitigation or measurement (Chzhen et al., 2020; Miroshnikov et al., 2020; 2021; Kallus & Zhou, 2019; Barata et al.). Robust fairness is also well studied (Mehrotra & Vishnoi, 2022; Ma et al., 2022; Chai & Wang, 2022; An et al., 2022; Giguere et al., 2022; Jiang et al., 2023), such as under distribution shift and with limited sensitive attributes.

**Metrics for Demographic Parity.** Metrics for the violation of demographic parity are proposed to evaluate the demographic parity. The widely used metrics are $\Delta DP_{continuous}$ (short for $\Delta DP_c$), which is calculated by the predictive probabilities (*continuous*), and $\Delta DP_{binary}^t$ (short for $\Delta DP_b^t$), which is calculated by the binary prediction (*binary*) with respect to a threshold $t$. Here we present their formal definitions. $\Delta DP_b^t$ (Dai & Wang, 2021; Creager et al., 2019; Edwards & Storkey, 2015; Kamishima et al., 2012) is the difference between the proportion of the positive prediction between different groups, which uses the difference of the average of the binary prediction between different groups. $\Delta DP_c$ (Chuang & Mroueh, 2020; Zemel et al., 2013) measures the difference in the mean of the predictive probability, which uses the difference in the average of the predictive probability between different groups. In addition, the metric based on Wasserstein distance for demographic parity also is proposed in (Jiang et al., 2020). The authors (Jiang et al., 2020) directly propose strong demographic parity on predictive probability using Wasserstein distance. Instead, in this paper, we start with the proposed two criteria (i.e., sufficiency, and fidelity), and then design fairness metrics as the distance of probability density function (PDF) and cumulative distribution function (CDF). Furthermore, we analyze that the proposed fairness metrics are actually some version of Wasserstein distance. The fairness metric for continuous sensitive attributes with Kernel Density Estimation (KDE) has also been proposed in works (Mary et al., 2019; Grari et al., 2020; Jiang et al., 2022b).

## 8 Conclusion

In this paper, we rethink the rationale of the widely adopted fairness metric $\Delta DP$ and propose two metrics for demographic parity with sufficiency and fidelity. Specifically, we first point out that the existing fairness metric is not the necessary and sufficient condition for demographic parity. We propose two fundamental criteria for fairness measurement design. Further, we propose two fairness metrics, namely ABPC and ABCC, which satisfy the proposed criteria with theoretical and experimental justification. Finally, we re-evaluate the three standard baselines on tabular and image datasets considering all fairness metrics of demographic parity. We believe the proposed metrics would contribute to the fairness of academic and industrial communities by properly evaluating the performance of fair models and suggesting a way to model development.

## Acknowledgements

We are grateful for the detailed reviews from the reviewers. Their thoughtful comments and suggestions were invaluable in strengthening this paper. We also would like to sincerely thank everyone who has provided their generous feedback for this work. This work was supported in part by National Science Foundation (NSF) IIS-1900990, IIS-1939716, IIS-2239257, and NSF IIS-2224843.

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

## A    Proof of Theorem 2.1

Firstly, we provide the relation between binary prediction $\hat{y}^t$ and the original predictive probability $\tilde{y}$, i.e., the original predictive probability $\tilde{y}$ is the mean of binary prediction $\hat{y}^t$ with uniform-distribution threshold $t$:

$$\int_0^1 \hat{y}_n^t \mathrm{d}t = \int_0^1 \mathbb{1}_{\geq t}[\tilde{y}_n]\mathrm{d}t = \int_0^{\tilde{y}_n} \mathrm{d}t = \tilde{y}_n, \tag{5}$$

Furthermore, we can derive the relation between these two fairness measurements based on the definition of $\Delta DP_c$ and $\Delta DP_b^t$ in Equation (1) and Equation (2):

$$\begin{aligned}
\int_0^1 \Delta DP_b^t \mathrm{d}t &= \int_0^1 \left| \frac{\sum_{n\in\mathcal{S}_0} \hat{y}_n^t}{N_0} - \frac{\sum_{n\in\mathcal{S}_1} \hat{y}_n^t}{N_1} \right| \mathrm{d}t \\
&\geq \left| \frac{\int_0^1 \sum_{n\in\mathcal{S}_0} \hat{y}_n}{N_0} - \frac{\sum_{n\in\mathcal{S}_1} \hat{y}_n}{N_1} \mathrm{d}t \right| \\
&= \left| \sum_{n\in\mathcal{S}_0} \frac{\int_0^1 \hat{y}_n \mathrm{d}t}{N_0} - \sum_{n\in\mathcal{S}_1} \frac{\int_0^1 \hat{y}_n \mathrm{d}t}{N_1} \right| \\
&= \left| \frac{\sum_{n\in\mathcal{S}_0} \tilde{y}_n}{N_0} - \frac{\sum_{n\in\mathcal{S}_1} \tilde{y}_n}{N_1} \right| = \Delta DP_c.
\end{aligned} \tag{6}$$

Note that if we set threshold as $t = 0$, it is easy to obtain that $\Delta DP_b^0 = \left| \frac{\sum_{n\in\mathcal{S}_0} \hat{y}_n^0}{N_0} - \frac{\sum_{n\in\mathcal{S}_1} \hat{y}_n^0}{N_1} \right| = 0 \leq \Delta DP_c$.

Additionally, according to Rolle's theorem (Ballantine & Roberts, 2002) and Equation (6), there exist certain threshold $t_0$ so that the following equation holds

$$\Delta DP_b^{t_0} = \int_0^1 \Delta DP_b^t \mathrm{d}t \geq \Delta DP_c, \tag{7}$$

Considering the continuity of fairness measurement $\Delta DP_b^t$ over threshold $t$, there exists threshold $t^* \in [0, t_0]$ so that $\Delta DP_b^{t^*} = \Delta DP_c$ holds, which completes the proof.

## B    More Discussion on Insufficiency of $\Delta DP$

**Proposition B.1** $\Delta DP_c^t = 0$ and $\Delta DP_b = 0$ are necessary but insufficient conditions for demographic parity, which requires that the prediction should be independent of the sensitive attributes.

If the predictive values satisfy demographic parity, the predictive probability should be independent of the sensitive attributes. Obviously, the distributions of the predictive probability of different groups follow the identical distribution, thus, the mean of the predictive probability of different groups should be the same. In practice, if the number of instances is large enough, $\Delta DP_b = 0$ and $\Delta_b^t = 0$ for any threshold hold. On the contrary, $\Delta DP_c = 0$ or $\Delta DP_b^t = 0$ for a certain threshold will not guarantee that the predictive probability of different groups follows the identical distribution. Thus we obtain that $\Delta DP_c = 0$ or $\Delta DP_b^t = 0$ is necessary but not sufficient conditions for demographic parity. This proposition illustrates that zero-value fairness measurements $\Delta DP_b^t = 0$ and $\Delta DP_c = 0$ can not guarantee that the model prediction and sensitive attributes are independent.

## C    Proof of Properties of ABPC and of ABCC

Firstly, it is easy to check the continuity condition since the PDF and CDF estimation are continuous over model prediction, and our proposed bias metrics ABPC and ABCC are also continuous w.r.t. the PDF and CDF estimation. As for the invariance over invertible transformation $T$, supposed the PDF $f_0(x)$ and $f_1(x)$ represent the PDF of model prediction for different groups, and the transformed prediction PDF is $\hat{f}_0(z)$

and $\hat{f}_1(z)$ with $z = T(x)$, note that the transformation $T$ is invertible, according to probability theory, the relation between $f_0(x)$ and $\hat{f}_0(z)$ satisfies $f_0(x)\mathrm{d}x = \hat{f}_0(z)\mathrm{z}$. Therefore, we have

$$\mathsf{ABPC}_z = \int_0^1 |\hat{f}_0(z) - \hat{f}_1(z)|\mathrm{d}z = \int_0^1 |f_0(x) - f_1(x)|\mathrm{d}x = \mathsf{ABPC}_x;$$

Lastly, we consider the necessary and sufficient conditions on demographic parity. It is easy to obtain that demographic parity represents the independent prediction w.r.t. sensitive attributes. Hence, ABPC and ABCC are zero, and vice versa.

## D  The Relation between ABPC and $\Delta DP_c$

Based on the definition of ABPC, we have

$$
\begin{aligned}
\mathsf{ABPC} &= \int_0^1 |f_0(x) - f_1(x)|\mathrm{d}x & (8) \\
&\geq \int_0^1 |xf_0(x) - xf_1(x)|\mathrm{d}x \\
&\geq \left| \int_0^1 xf_0(x)\mathrm{d}x - \int_0^1 xf_1(x)\mathrm{d}x \right| & (9) \\
&= \Delta DP_c.
\end{aligned}
$$

i.e., the bias metric ABPC is rigorously larger than $\Delta DP_c$, which conquers the drawback of the insufficient condition of demographic parity.

## E  The Relation between ABPC and ABCC

Based on the definition of Wasserstein distance with cost function $c_0(x, y) = 2 \cdot \mathbb{1}(x \neq y)$, we have

$$
\begin{aligned}
&\frac{1}{2}W_{c_0}(f_0(x), f_1(x)) \\
&= \inf_{\gamma \in \Gamma(f_0(x), f_1(x))} \int_{[0,1]^2} \mathbb{1}(x \neq y)\gamma(x, y)\mathrm{d}x\mathrm{d}y \\
&= \inf_{\gamma \in \Gamma(f_0(x), f_1(x))} \int_{[0,1]^2} \big(1 - \mathbb{1}(x = y)\big)\gamma(x, y)\mathrm{d}x\mathrm{d}y \\
&= 1 - \int_0^1 \min\{f_0(x), f_1(x)\}\mathrm{d}x \\
&= \frac{1}{2}\int_0^1 |f_0(x) - f_1(x)|\mathrm{d}x & (10) \\
&= \frac{1}{2}\mathsf{ABPC}.
\end{aligned}
$$

According to work (Shorack & Wellner, 2009), when the PDF of prediction has a limited expectation, we take the following equation:

$$
\begin{aligned}
\mathsf{ABCC} &= \int_0^1 |F_0(x) - F_1(x)|\mathrm{d}x \\
&= \int_0^1 |F_0^{-1}(t) - F_1^{-1}(t)|\mathrm{d}t & (11) \\
&= W_1\big(f_0(x), f_1(x)\big).
\end{aligned}
$$

where $F_0^{-1}(t)$ represents the inverse function of the original CDF $F_0(x)$. Therefore, the proposed two metrics ABPC and ABCC are both the distance for these two PDFs for different groups except the distance metrics.

## F  Proof of Estimation Tractability on ABPC and ABCC

According to the non-parametric estimation theory (Sampson & Guttorp, 1992; Jiang et al., 2022b), the estimation error for PDF satisfies $Error_{pdf} = \mathbf{E}_x[||f(x) - \hat{f}(x)||^2] = O(N^{-\frac{4}{5}})$ for kernel density estimator. Hence, for bias metric ABPC estimation error, we have

$$
\begin{aligned}
Error_{\mathsf{ABPC}} &= \mathbf{E}_{\mathcal{D}}\big[|\mathsf{ABPC} - \hat{\mathsf{ABPC}}|^2\big] \\
&= \mathbf{E}_x\big[|\int_0^1 |f_0(x) - f_1(x)|\mathrm{d}x - \int_0^1 |\hat{f}_0(x) - \hat{f}_1(x)|\mathrm{d}x|^2\big] \\
&\overset{(a)}{\leq} \mathbf{E}_x\big[|\int_0^1 |f_0(x) - \hat{f}_0(x)|\mathrm{d}x + \int_0^1 |f_1(x) - \hat{f}_1(x)|\mathrm{d}x|^2\big] \\
&= 2\Big[\big(\mathbf{E}_x[\int_0^1 |f_0(x) - \hat{f}_0(x)|\mathrm{d}x]\big)^2 + \big(\mathbf{E}_x[\int_0^1 |f_1(x) - \hat{f}_1(x)|\mathrm{d}x]\big)^2\Big] \\
&\overset{(b)}{\leq} 2\Big[\mathbf{E}_x[\int_0^1 |f_0(x) - \hat{f}_0(x)|^2\mathrm{d}x] + \mathbf{E}_x[\int_0^1 |f_1(x) - \hat{f}_1(x)|^2\mathrm{d}x]\Big] \\
&= O(N^{-\frac{4}{5}}).
\end{aligned}
$$

where inequality (a) holds due to absolute inequality, and inequality (b) holds based on Cauchy-Schwarz inequality for the integration version. For the number of samples $N = O(\delta^{-\frac{5}{4}}\epsilon^{-\frac{5}{2}})$, we have

$$
\begin{aligned}
\mathbb{P}(|\mathsf{ABPC} - \hat{\mathsf{ABPC}}| < \epsilon) &= 1 - \mathbb{P}(|\mathsf{ABPC} - \hat{\mathsf{ABPC}}|^2 \geq \epsilon^2) \\
&\overset{(c)}{\geq} 1 - \frac{\mathbf{E}_{\mathcal{D}}\big[|\mathsf{ABPC} - \hat{\mathsf{ABPC}}|^2\big]}{\epsilon^2} \\
&= 1 - \delta,
\end{aligned}
\tag{12}
$$

where inequality (c) holds due to Markov chain inequality. In other words, for the sufficient number of samples $N = O(\delta^{-\frac{5}{4}}\epsilon^{-\frac{5}{2}})$, $|\mathsf{ABPC} - \hat{\mathsf{ABPC}}| < \epsilon$ holds with at least probability $1 - \delta$.

Similarly, for CDF estimation, based on the central limit theorem, $\sqrt{n}(\hat{F}(x) - F(x))$ has asymptotically normal distribution, i.e., $Error_{cdf} = \mathbf{E}_x[||F(x) - \hat{F}(x)||^2] = O(N^{-1})$. Therefore, similar to the derivation of PDF, we can obtain $Error_{\mathsf{ABPC}} = O(N^{-1})$. For the number of samples $N = O(\delta^{-1}\epsilon^{-2})$, we can also find that $|\mathsf{ABCC} - \hat{\mathsf{ABCC}}| < \epsilon$ holds with at least probability $1 - \delta$.

## G  Experimental Examination on Adult Dataset

In this appendix, we provide an experimental examination of the Adult dataset in Figure 7, which is the same as Figure 2. We can observe that the PDF and CDF of biased MLP are obviously different, illustrating that the machine learning model is biased to gender. Debiased MLP achieves a much lower $\Delta DP_c$ than the biased MLP, however, the PDFs of different groups are different even though the "mean" of them are almost the same. The bottom figures show that $\Delta DP_b^t$ is the difference between the CDF lines while the threshold $t = 0.5$. The results are also in line with the finding in Section 3.2.

## H  Re-evaluate the Fairness on Image Data

In this appendix, we conduct an experiment on the CelebA image dataset to re-evaluate the performance of the fair model in Table 2. The CelebA face attributes dataset (Liu et al., 2015) contains over 200,000 face images, where each image has 40 human-labeled attributes. Among the attributes, we select Attractive as a binary classification task and consider gender as the sensitive attribute. The results are presented in Table 2. The results show a similar finding with the tabular dataset, demonstrating that 1): REG method always achieves a lower $\Delta DP_c$ but a relatively high $\Delta DP_b^t$. 2): ADV is a more promising fair model to achieve demographic parity. 3): The inherent tension between fairness and accuracy is underestimated.

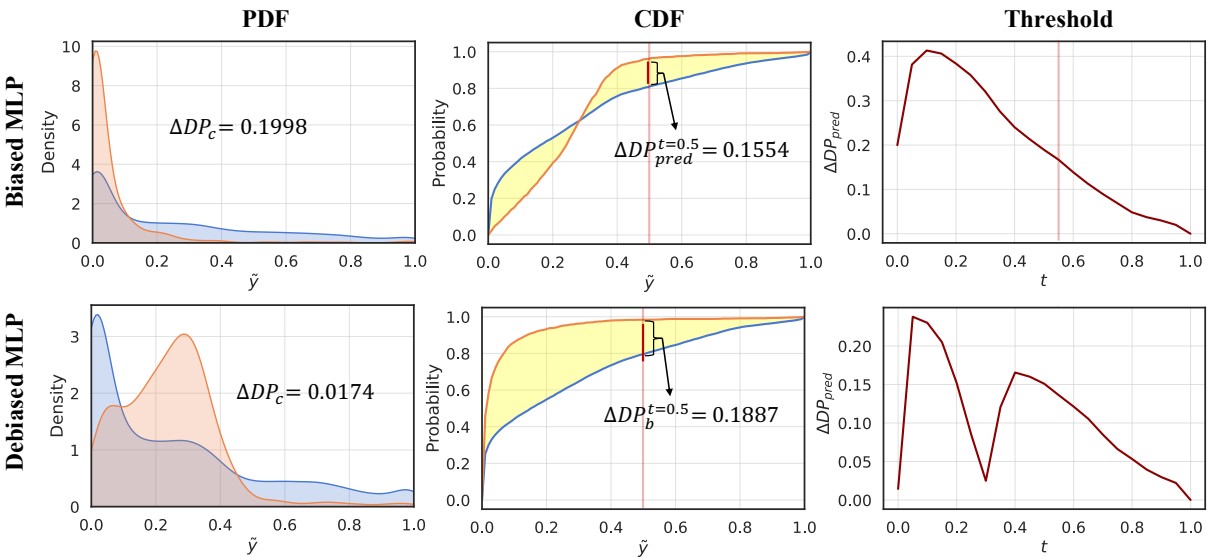

Figure 7: The empirical PDF and CDF of the predictive probability over different groups (i.e., male, female) on Adult dataset. *Left*: Biased MLP model. *Right*: Debiased MLP model with $\Delta DP_c$ as regularization.

Table 2: The fairness performance on image dataset. ↑ represents the accuracy improvement compared to MLP. A higher accuracy metric indicates better performance. ↓ represents the improvement of fairness compared to MLP. A lower fairness metric indicates better fairness.

| | | Accuracy | | | Fairness | | | | | | | |
|---|---|---|---|---|---|---|---|---|---|---|---|---|
| | | Acc(%) | ↑ | AP(%) | ↑ | $\Delta DP_b^t(\%)$ | ↓ | $\Delta DP_c(\%)$ | ↓ | ABPC(%) | ↓ | ABCC(%) | ↓ |
| Age | MLP | **79.12** | — | **88.06** | — | 44.17 | — | 44.11 | — | 46.70 | — | 44.11 | — |
| | REG | 66.28 | -16.22 | 86.08 | -2.24 | **2.10** | 99.95 | 14.02 | 68.21 | 65.69 | -40.66 | 17.21 | 60.98 |
| | ADV | 59.48 | 24.82 | 63.93 | 27.40. | 12.31 | 72.13 | 12.15 | 72.46 | 16.73 | 64.18 | 12.15 | 72.46 |
| Gender | MLP | **79.12** | — | **88.06** | — | 42.21 | — | 41.87 | — | 43.61 | — | 41.87 | — |
| | REG | 77.82 | -1.64 | 72.63 | -17.52 | 30.03 | 28.86 | 02.00 | 95.22 | 19.36 | 55.61 | 22.42 | 46.45 |
| | ADV | 65.25 | -17.53 | 71.20 | -19.15 | **01.91** | 95.48 | **01.91** | 95.44 | 04.00 | 90.83 | 01.91 | 95.44 |

## I Wasserstein Distance Introduction

The Wasserstein distance (Rüschendorf, 1985) has already been adopted in machine learning due to the power of measuring the difference between two distributions. In the fairness community, the transport problem measures the difference in predictive probability for different groups. Formally, suppose the predictive probability distributions for different groups are $f_0(x)$ and $f_1(x)$, and the cost function moving from $x$ to $y$ is $c(x,y)$, the transportation problem is given by $\gamma^*(x,y) = \arg \inf_{\gamma \in \Gamma(f_0,f_1)} \int c(x,y)\gamma(x,y)\mathrm{d}x\mathrm{d}y$, where distribution set $\Gamma(f_0, f_1) = \left\{\gamma > 0, \int \gamma(x,y)\mathrm{d}y = f_0(x), \int \gamma(x,y)\mathrm{d}x = f_1(y)\right\}$ is the collection of all possible transportation plan, i.e., joint distribution with margin distribution $f_0$ and $f_1$. The optimal transportation plan always exists and is defined as $\gamma^*(x,y)$.

The $p^{th}$ Wasserstein distance is a special case of the optimal transport problem with cost function $c(x,y) = |x - y|^p$. The formal definition is given by

$$W_p(f_0, f_1) = \left(\inf_{\gamma \in \Gamma(f_0,f_1)} \int |x - y|^p \gamma(x,y)\mathrm{d}x\mathrm{d}y\right)^{\frac{1}{p}}. \tag{13}$$

**Algorithm 1** Python code of ABPC

```python
def ABPC( y_pred, y_gt, z_values, bw_method = "scott",
    sample_n = 5000 ):
    # y_pred: predicted values (continuous).
    # y_gt: ground truth label (binary).
    # z_values: sensitive attributes (binary).
    # bw_method: the method for estimator bandwidth
    # sample_n: the number of sample points to use for
        integration of the area between the PDFs.

    y_pred = y_pred.ravel()
    y_gt = y_gt.ravel()
    z_values = z_values.ravel()

    y_pre_1 = y_pred[z_values == 1]
    y_pre_0 = y_pred[z_values == 0]

    # KDE PDF
    kde0 = gaussian_kde(y_pre_0, bw_method = bw_method)
    kde1 = gaussian_kde(y_pre_1, bw_method = bw_method)

    # integration
    x = np.linspace(0, 1, sample_n)
    kde1_x = kde1(x)
    kde0_x = kde0(x)
    abpc = np.trapz(np.abs(kde0_x - kde1_x), x)

    return abpc
```

**Algorithm 2** Python code of ABCC

```python
def ABCC( y_pred, y_gt, z_values, sample_n = 10000 ):
    # y_pred: predicted values (continuous).
    # y_gt: ground truth label (binary).
    # z_values: sensitive attributes (binary).
    # sample_n: The number of sample points to use for
        integration of the area between the CDFs.

    y_pred = y_pred.ravel()
    y_gt = y_gt.ravel()
    z_values = z_values.ravel()

    y_pre_1 = y_pred[z_values == 1]
    y_pre_0 = y_pred[z_values == 0]

    # empirical CDF
    ecdf0 = ECDF(y_pre_0)
    ecdf1 = ECDF(y_pre_1)

    # integration
    x = np.linspace(0, 1, sample_n)
    ecdf0_x = ecdf0(x)
    ecdf1_x = ecdf1(x)
    abcc = np.trapz(np.abs(ecdf0_x - ecdf1_x), x)

    return abcc
```

# J   Python Code for the Proposed Metrics

In this appendix, we provide the python code for our proposed two metrics, ABPC and ABCC. The provided codes show that our proposed metrics are practical and easy to compute.

The numbers $5,000$ and $10,000$ mentioned for ABPC and ABCC are not the number of data samples. Instead, they refer to the number of points used in the integration process for both metrics. Both ABPC and ABCC are based on estimated functions with a range of $[0, 1]$. The "sample_n" parameter represents the number of points within the range $[0, 1]$ used for integration, and the integration is computed using these points. Thus, we can set "sample_n" to a relatively large number. The default values of $5,000$ and $10,000$ are not strict requirements. In our experiments, we found that a "sample_n" larger than 100 should be sufficient for accurate results.

