# OpenReview forum: "Retiring $\Delta \text{DP}$: New Distribution-Level Metrics for Demographic Parity"
_TMLR — Accepted by TMLR_

### Review · Reviewer_YqvS · 2023-03-08

**Summary Of Contributions:**

Demographic Parity (DP) is a commonly used metric for evaluating the fairness of the models. This work argues that DP has two shortcomings - 1) 0 DP does not imply 0 violation of demographic parity 2) DP can vary based on the threshold chosen for classification. This paper proposes two new metrics Area Between Probability density function Curves (ABPC) and Area Between Cumulative density function Curves (ABCC) to deal with these shortcomings. These metric measure the distance between the distributions instead of the average values. Then, they evaluate a few machine learning models trained with different fairness regularizations and compare the values of different fairness metrics.

**Audience:**

Yes

**Claims And Evidence:**

Yes

**Requested Changes:**

- This paper studies DP and points out concerns with this measure. This has already been done by previous works. For example, [1] already pointed out how DP is not a sufficient metric and introduced the Equalized odds metric. Can the authors please comment on what different metrics have been introduced to improve over DP’s shortcomings and how their method is different from existing ones?

- On Page 1 in the introduction, the first line containing DP mentions “In this paper, we rethink the rationale of ∆DP”. It would be useful for the reader to get a 2 line introduction to the metric first and then mention that this paper rethinks that.

- DP and the new proposed metrics don’t not look at the feature distribution at all. Can the authors include a discussion on metrics which also depend on feature distribution and what are the tradeoffs between two two types of metrics?

- In [2], the authors argue that some natural fairness conditions can not be all satisfied simultaneously. Can the authors comment on their metrics in this regard that what conditions are not satisfied by the proposed metrics?

- In section 5.5.1, it is not clear mutual information is calculated between which variables.

- In figure 5, the authors claim that DP_c is not a good metric since it is lower for plot b compared to c. How about DP_b? Is that a good metric for this case?

- Do the authors have some intuition on why ADV performs better with respect to their metrics?

- The authors comment on how different fairness regularization methods work w.r.t the proposed metrics in section 6.3. Is there some way to verify that they are actually better?

- Is there a way to directly optimize the proposed metrics?

Typos:

- Preliminaries section: where n represents the -> where N represents the
- Figure 2 description: not clear if talking about the top or bottom figure in the first two lines. The figure mentions Left and Right instead of Top and Bottom. The y label of the rightmost plot is cut.

[1] Equality of Opportunity in Supervised Learning

[2] Inherent Trade-Offs in the Fair Determination of Risk Scores

**Strengths And Weaknesses:**

- The DP metric is a very common metric used in fairness literature and building on top of that and highlighting its shortcomings can be interesting for the community.

- The paper is overall well written.

- This notion of distance between distributions as a stronger form of DP has already been introduced in Jiang et. al. And, they also consider the same distribution distance that this work considers. But, as the authors mention, they derive these metrics starting from desired conditions.

---

> ### Author Response · Authors · 2023-03-15
> **Response to Reviewer YqvS [1/3]**
>
> Dear Reviewer YqvS,
>
> Thank you for reviewing our paper and providing valuable feedback. We are glad that you found the presentation and significance satisfactory. We have taken your constructive comments into account and made the according revisions to our revised paper. To help you easily identify the changes made in response to your concerns, we highlighted the revisions in an ${\color{orange}orange}$ note box within the margin of our revised paper.
>
> **1.Can the authors please comment on what different metrics have been introduced to improve over DP’s shortcomings and how their method is different from existing ones?**
>
> Thank you for your constructive comment! Instead of invalidating the fairness definition of demographic parity, we assert that the currently widely-used metrics for demographic parity (i.e., $\Delta DP_b$ and $\Delta DP_c$) cannot precisely reflect the violation of demographic parity. In our paper, to accurately measure the violation of demographic parity, we propose two new and reasonable metrics, ABPC and ABCC. We have clarified this point at the end of Section 1 Introduction on Page 2.
>
>
> **2. It would be useful for the reader to get a 2 line introduction to the metric first and then mention that this paper rethinks that.**
>
> Thank you for your constructive comment. We have revised this section of the paper accordingly. We revised the first line introducing $\Delta DP$ to *In this paper, we focus on the measurement of demographic parity, $\Delta DP$, which requires the predictions of a machine learning model should not be dependent on sensitive attributes~\citep{menon2018cost,ustun2019fairness,kamishima2012fairness}.*
>
>
> **3. DP and the new proposed metrics don’t not look at the feature distribution at all. Can the authors include a discussion on metrics which also depend on feature distribution and what are the tradeoffs between two two types of metrics?**
>
> Thank you for your insightful comment! To the best of our knowledge, we have not come across any fairness metric that involves feature distribution. We also reviewed the existing literature [3][4][5] again on fairness definitions and did not find such metrics.
>
> As you suggested, a fairness metric involving feature distribution would be interesting and promising, as it could define fairness by focusing on the "debiasing" effect of the model. We appreciate your suggestion and will consider it for future research.
>
>
> **4. In [2], the authors argue that some natural fairness conditions can not be all satisfied simultaneously. Can the authors comment on their metrics in this regard that what conditions are not satisfied by the proposed metrics?**
>
> Thank you for your thoughtful feedback. We agree that some fairness conditions cannot be satisfied simultaneously, such as Demographic Parity and Equality of Opportunity. In our paper, we only focus on proposing new and reasonable metrics to measure the violation of demographic parity. Both proposed metrics are for demographic parity. The proposed metrics $ABPC/ABCC$ will be $0$ simultaneously if and only if the prediction probabilities $\tilde{Y}$ are independent of sensitive attributes $S$. Since our proposed ABPC and APCC are more general and stronger metrics than $\Delta DP$, demographic parity and equality of opportunity cannot hold simultaneously as well, except in trivial cases.
>
>
> **5. In section 5.5.1, it is not clear mutual information is calculated between which variables.**
>
> Thank you for your valuable suggestion. We clarify this point as follows: Mutual information is calculated based on prediction probability and conditional prediction probability, i.e., $MI(\tilde{Y};S)=H(\tilde{Y}) - H(\tilde{Y}|S)=H(\tilde{Y}) - \mathbb{P}(S=0)H(\tilde{Y}|S=0)-\mathbb{P}(S=1)H(\tilde{Y}|S=1)$, where the entropy function $H(\tilde{Y})=\int_{0}^{1} f_{\tilde{Y}}(\tilde{y})\mathrm{d}\tilde{y}$, and $H(\tilde{Y}|S=i)=\int_{0}^{1} f_{\tilde{Y}|S=i}(\tilde{y})\mathrm{d}\tilde{y}$ for $i=0,1$. Mutual information between model prediction $\tilde{Y}\in[0,1]$ and sensitive attribute $S\in\{0,1\}$ can be adopted to measure the independence. Compared with the definitions of ABPC and ABCC, the calculation of MI is also based on the estimated (conditional) prediction probability, as shown in Section 5.3. The advantage of our proposed ABPC and ABCC is the lower metric relative estimation error. We have revised the paper in Section 5.5.1 accordingly.

---

> ### Author Response · Authors · 2023-03-15
> **Response to Reviewer YqvS [2/3]**
>
>
> **6. In figure 5, the authors claim that DP_c is not a good metric since it is lower for plot b compared to c. How about DP_b? Is that a good metric for this case?**
>
> Yes, $\Delta DP_b$ was able to measure the violation of Model 2 in Figure 5 on the Adult dataset, where $\Delta DP_b=0.1554$ indicates that the model is unfair. However, for Model 2 in Figure 1 on the ACS-Income dataset, both $\Delta DP_b$ and $\Delta DP_c$ failed to evaluate the demographic parity violation. Therefore, $\Delta DP_b$ cannot consistently evaluate models across different datasets, which is another experimental evidence of its failure to measure the violation of demographic parity.
>
> Regarding the model side, this may be due to the fact that current fairness methods often result in trivial solutions when using $\Delta DP_b$ and $\Delta DP_c$ as measurements.
>
> We have also revised the experimental analysis in Section 6.3 accordingly.
>
>
>
> **7. Do the authors have some intuition on why ADV performs better with respect to their metrics?**
> We thank you for the thoughtful feedback. The reason why ADV performs better using our proposed metrics is twofold. First, adversarial learning can intuitively and theoretically ensure that the predicted probabilities (a continuous value) are independent of the sensitive attributes. This is supported by both theoretical and empirical evidence. Second, our two proposed metrics are designed to be 0 only when the predicted probabilities are fully independent of the sensitive attributes. This provides a more strict and precise measure of demographic parity, which may make it easier for the ADV method to achieve better performance on this metric.
>
> On the other hand, the REG method in our paper uses $\Delta DP_c$ as a fairness regularization term (as well as a surrogate loss function for $\Delta DP_c$). By using the difference of the average predicted probabilities for demographic groups, it cannot ensure that the predicted probabilities are fully independent of the sensitive attributes. Thus, it may achieve secondary performance on our proposed metric.
>
> The experimental observation that ADV performs better with respect to their metrics somehow shows that measuring demographic parity from the distribution of predicted probabilities can provide more insight into the behavior of existing fairness methods.
>
> We thank you for the thoughtful comment again, and we have also revised the paper in Section 6.3.
>
>
>
> **8. The authors comment on how different fairness regularization methods work w.r.t the proposed metrics in section 6.3. Is there some way to verify that they are actually better?**
>
> Thank you for your thoughtful feedback. Thank you for your thoughtful feedback. Our proposed metrics offer a more precise measure of demographic parity violation because they directly assess the independence between prediction probabilities and sensitive attributes, which can be regarded as the 'ground truth' of demographic parity in real-world applications. Our proposed metric uses the difference in the distribution of the prediction probabilities to precisely measure the independence between prediction probabilities and sensitive attributes. In this sense, our proposed metric is able to reflect the degree of independence between the two.
>
>
>
>
> **9. Is there a way to directly optimize the proposed metrics?**
>
> Thank you for your thoughtful suggestion. The proposed ABPC and ABCC metrics are both differentiable, which enables their direct optimization. This property arises from the fact that the (conditional) prediction probability density can be estimated using kernel density estimation (KDE), i.e., $\tilde{f}{i}(x)=\frac{1}{|\mathcal{S}i|h}\sum{n\in\mathcal{S}i}K(\frac{x-\tilde{y}{n}}{h})$, where $K(x)$ is a smoothing Gaussian kernel function, $\tilde{y}_{n}$ is the model prediction for the $n$-th sample, and $h$ is a pre-defined bandwidth. Note that ABPC is differentiable with respect to (conditional) prediction probability density, and (conditional) prediction probability density is differentiable with respect to model prediction and model parameters. Therefore, ABPC can be directly optimized. Similarly, ABCC can also be directly optimized. We would like to clarify that this paper primarily focuses on the development of fairness metrics, and the development of bias mitigation methods for ABPC and ABCC is left for future work.

---

> ### Author Response · Authors · 2023-03-15
> **Response to Reviewer YqvS [3/3]**
>
> **Typos**
>
> We thank you for your careful reading and pointing out the typos.  We revised paper accorddingly as follows:
>
> 1. In the Preliminaries section, we changed "where n represents the" to "where N represents the" to maintain consistency in notation.
> 2. For Figure 2, we apologize for the confusion. We have clarified the description in the caption to accurately reference the subfigures as Top and Bottom.
> 3. We also fixed the y-label of the rightmost plot to ensure it is not cut off and is displayed properly.
> 4. We have also carefully gone through the entire paper and fixed the typos we found.
>
>
>
> We sincerely thank you once again for your time and feedback. We hope our response has addressed your concerns, and we are glad to answer any further questions you may have.
>
>
> Thanks,\
> Authors
>
>
> **Reference**\
> [1] Equality of Opportunity in Supervised Learning\
> [2] Inherent Trade-Offs in the Fair Determination of Risk Scores\
> [3] Garg, Pratyush, John Villasenor, and Virginia Foggo. "Fairness metrics: A comparative analysis." In 2020 IEEE International Conference on Big Data (Big Data), pp. 3662-3666. IEEE, 2020.\
> [4] Verma, Sahil, and Julia Rubin. "Fairness definitions explained." In Proceedings of the international workshop on software fairness, pp. 1-7. 2018.\
> [5] Mitchell, Shira, Eric Potash, Solon Barocas, Alexander D'Amour, and Kristian Lum. "Algorithmic fairness: Choices, assumptions, and definitions." Annual Review of Statistics and Its Application 8 (2021): 141-163.\

---

### Review · Reviewer_n71L · 2023-03-08

**Summary Of Contributions:**

This paper considers metrics for demographic parity in the classification task.
The contribution of this paper includes the following points:
- This paper points out issues with the standard metrics, $\Delta DP_b$ and $\Delta DP_c$.
- To resolve the issues, this paper proposes new metrics referred to as ABPC and ABCC.
- Using new metrics ABPC adn ABCC, this paper re-evaluated the fairness performance of existing methods with real-worlds datasets.

**Audience:**

Yes

**Broader Impact Concerns:**

I have no concerns about ethical aspects.

**Claims And Evidence:**

Yes

**Requested Changes:**

I would like you to check if there is any misunderstanding about the points I mentioned as weakness.
If there is no misunderstanding, please address it.

**Strengths And Weaknesses:**

Strengths:
- The code of the experiment is published in a form that is easy to reproduce.
- The paper is well organized and comfortable to read.
- The claims of the paper are supported by convincing theoretical and experimental evidence.

Weaknesses:

There is a lack of explanation as to when and in what situations the existing metrics ($\Delta DP_b$ and $\Delta DP_c$) are problematic,
especially in the introduction.
This paper focuses on models that output predictive probability $\tilde{y} \in [0,1]$ once and determines the label $\hat{y}$ using an appropriate threshold value $t$, i.e., $\hat{y} = \mathbf{1} [\tilde{y} \geq t]$.
In my understanding,
if $t$ is fixed and if the predicted label $\hat{y}$ alone is used in the downstream task,
$\Delta DP_b^t$ is a "rational" fairness metric,
i.e.,
$\Delta DP_b^t = 0$ (asymptotically) iff $\hat{y}$ is independent of the sensitive attributes.
In other words, the issues pointed out by the authors arise only when different threshold values $t$ are used depending on the downstream tasks; otherwise, the conventional metrics are sufficient.
My point is not to deny the intrinsic value of this study, but the way the paper is now written, it seems to me that this is overstating the magnitude of the problem of existing methods and the scope of application of the proposed method.

---

> ### Author Response · Authors · 2023-03-15
> **Response to Reviewer n71L**
>
> Dear Reviewer n71L,
>
> We thank you for taking the time to review our paper. We are glad that you found the presentation, theoretical and experimental support, and reproducibility to be satisfactory.  We also thank you for your constructive feedback of our method. We have revised the paper accordingly and highlighted the revision to address your concerns in the ${\color{purple}purple}$ note box on the margin of our revised paper.
>
> **1. When and in what situations the existing metrics ( $\Delta DP_b$ and $\Delta DP_c$ ) are problematic**
>
> Existing metrics, $\Delta DP_b$ and $\Delta DP_c$, have two major limitations:
>
> 1. These two metrics are not suitable in many real-world applications, where the classifier's threshold needs to be modified. Changing the threshold is a common practice for decision-making but can violate demographic parity if the model is evaluated using $\Delta DP$ metric. There are several examples to illustrate that changing the threshold in the downstream task. For instance, in college admissions, the number of admissions and applicants can vary from year to year, requiring the threshold for decision-making to be adjusted accordingly. Similarly, in AI-assisted medical diagnosis, doctors may adjust the threshold for diagnosing a disease based on the patient's family history. As demonstrated in [1], loading predictions also require a threshold for credit score.  However, if the model is evaluated using the $\Delta DP$ metric, demographic parity cannot be guaranteed if the threshold is changing.
> 2. Direct optimization over $\Delta DP_b$ and $\Delta DP_c$ for machine learning models can lead to trivial but unfair solutions. The current fairness methods usually result in trivial solutions if we use $\Delta DP_b$ and $\Delta DP_c$ as measurements, as shown in the results of Model 2 in Figures 1 and 5. The trivial solution may obtain a lower $\Delta DP_b$ and $\Delta DP_c$, but it is an unfair model. However, $\Delta DP_b$ and $\Delta DP_c$ have become the *de facto* measurements for the current fairness metric, as many previous works use them as fairness metrics (e.g., [2][3] use $\Delta DP_b$ and [4][5] use $\Delta DP_c$). This could mislead the development of fairness methods in terms of demographic parity.
>
> Thank you for your suggestion. We have revised the introduction in Page 2 accordingly.
>
>
> **2. if t is fixed and if the predicted label alone is used in the downstream task, $\Delta DP_b$ is a "rational" fairness metric, i.e., $\Delta DP_b=0$ (asymptotically) iff $\hat{y}$ is independent of the sensitive attributes.**
>
> Thank you for the great question! We agree that $\Delta DP_b=0$ (asymptotically) if and only if $\hat{y}$ is independent of the sensitive attributes since the conditional probability distribution $\hat{y}$ given sensitive attribute $S=0$ and $S=1$ are the same for the same ratio of positive samples across different demographic groups, i.e., $\Delta DP_b = \left| \frac{\sum_{n\in\mathcal{S}0} \hat{y}^{t}{n} }{N{0} } - \frac{\sum_{n\in\mathcal{S}1} \hat{y}^{t}{n} }{N{1} } \right|=0$. However, we argue that our proposed metrics are a stronger version of $\Delta DP_b$, especially for dynamic thresholds. In other words, when our proposed metric ABCC or ABPC is zero, $\hat{y}$ is independent of the sensitive attributes for any threshold. Conversely, $\Delta DP_b=0$ implies that $\hat{y}$ is independent of the sensitive attributes for specific thresholds. In a nutshell, our proposed fairness metrics are more general and stronger demographic parity metrics, which can be adopted in scenarios with dynamic thresholds.
> We have provided more discussion in the revised manuscript in Section 2.1.
>
>
>
> We sincerely thank you once again for your time and feedback. We hope our response has addressed your concerns, and we are glad to answer any further questions.
>
> Thanks,\
> Authors
>
>
> **Reference**\
> [1] Hardt, Moritz, Eric Price, and Nati Srebro. "Equality of opportunity in supervised learning." NeurIPS2016.\
> [2] Edwards, Harrison and Amos J. Storkey. “Censoring Representations with an Adversary.” ICLR2016.\
> [3] Creager, Elliot, David Madras, Jörn-Henrik Jacobsen, Marissa Weis, Kevin Swersky, Toniann Pitassi, and Richard Zemel. "Flexibly fair representation learning by disentanglement." ICML2019.\
> [4] Chuang, Ching-Yao, and Youssef Mroueh. "Fair Mixup: Fairness via Interpolation." ICLR2021.\
> [5] Zemel, Rich, Yu Wu, Kevin Swersky, Toni Pitassi, and Cynthia Dwork. "Learning fair representations." ICML2013.

---

### Review · Reviewer_tEuV · 2023-03-29

**Summary Of Contributions:**

This paper studies the metric of the fairness. The authors first show that the commonly used fairness metric $\Delta DP$ can not precisely measure the violation of demographic parity, and then propose two novel fairness metrics called ABCC and ABPC. The proposed metrics are: 1) necessary and sufficient to measure the demographic parity; 2) invariant with respect to monotone transformations of the distributions. The authors also re-evaluate existing fair models with the proposed metrics.

**Audience:**

Yes

**Claims And Evidence:**

Yes

**Requested Changes:**

1. The abstract is repeated twice.
2. In section 3.1, the paper may provide a toy example where the \Delta DP fails.
3. In section 5.2, the theoretical properies of ABPC and ABCC should be written as theorems with proofs.
4. In the last paragraph of section 5.3, "Similarly, ABCC is also can be directly optimized" ->  "Similarly, ABCC can also be directly optimized"
5. In Apendix B, $\Delta_b^t$ seems to be undefined.
6. The paper may provide some discussion on the issues presented in the weakness part.

**Strengths And Weaknesses:**

Strengths:

The fairness issue of machine learning methods is an important topic. The authors point out the drawback of the metric $\Delta DP$ and verify their claims both theoretically and empirically. The paper proposes two fundamental criteria for fairness measurement design and propose two fairness metrics which satisfy the proposed criteria.

Weakness:

1. The writing can be improved, the detail suggestions are presented in the "requested changes"
2. ABCC and ABPC cannot be computed directly. The paper may discuss the computational cost of the estimation of ABCC and ABPC.
3. According to lemma 5.2, the estimation of ABPC requires more samples than ABCC to guarantee the $\epsilon$ estimation error. But in the experiments, ABPC use 5k samples and ABCC use 10k samples. Why ABCC needs more samples for estimation in the experiments?

---

> ### Author Response · Authors · 2023-03-30
> **Response to Reviewer tEuV [1/2]**
>
> Dear Reviewer tEuV,
>
> We thank you for your time and effort in reviewing our paper. We appreciate the recognition of the importance of addressing fairness issues in machine learning and our efforts to identify drawbacks of existing metrics. We are grateful for your valuable feedback and suggestions regarding our paper. We have revised the paper accordingly and highlighted the revisions made to address your concerns in the ${\color{blue}blue}$ note box on the margin of our revised paper.
>
>
> ### Requested Changes:
> **1. The abstract is repeated twice.** \
> We appreciate you bringing this to us. We mistakenly updated the wrong version of our paper. We have corrected the issue by uploading a new version.
>
> **2. In section 3.1, the paper may provide a toy example where the \Delta DP fails.** \
> We thank you for the great suggestion! We have added two toy examples in Section 3.1 to further illustrate the two arguments.
>
> For *Augement 1: $\Delta DP=0$ is only a necessary but insufficient condition for demographic parity to hold.*, we present a toy example to illustrate it. Let's consider we have the model's prediction $\hat{y}=[0.4, 0.4, 0.4, 0.4, 0.5, 0.5, 0.5, 0.5, 0.5, 0.9]$, and the sensitive attribute $s=[0,0,0,0,1,1,1,1,1,0]$, We can see that the model is unfair since it tends to predict low values for samples with sensitive attribute $0$, while predicting high values for samples with sensitive attribute $1$, despite an outlier (the last sample). However, $\Delta DP_{c} = 0$, indicating a fair prediction. Thus, $\Delta DP_{c}$ may fail to measure fairness.
>
> For *Augment 2:Threshold Rules harm the measurement accuracy of $\Delta DP_{b}^{t}$*, we present the following toy example to illustrate it. Let's consider the we have the model's prediction $\hat{y}=[0.35, 0.45, 0.55, 0.65]$ and the sensitive attribute $s=[0,1,0,1]$, if we set the threshold to $0.5$ for classification, $\Delta DP_{b}^{0.5}=0$, indicating a fair prediction. However, if we set the threshold to $0.6$, $\Delta DP_{b}^{0.6}=0.67$, indication unfair prediction. Thus, with different thresholds, $\Delta DP_{b}^{t}$ may fail to measure fairness when the threshold changes.
>
>
>
>
> **3. In section 5.2, the theoretical properties of ABPC and ABCC should be written as theorems with proofs.** \
> Thanks for the constructive comments. We have revised the theoretical properties as theorems with proofs in Section 5.2.
>
>
> **4. In the last paragraph of section 5.3, "Similarly, ABCC is also can be directly optimized" -> "Similarly, ABCC can also be directly optimized"** \
> Thank you for carefully reading our paper. We have fixed the grammar error and proofread the paper to ensure readability.
>
>
>
> **5. In Appendix B, $\Delta_b^t$ seems to be undefined.** \
> We thank you for bringing this to us. $\Delta_b^t$ is typo. We have revised it as $\Delta DP_b^t$ in Appendix B.
>
>
> **6. The paper may provide some discussion on the issues presented in the weakness part.** \
> Thank you for your insightful feedback. We address your concerns about the weaknesses as follows and have also revised the paper based on your concerns:
> 1. We have added a discussion about the computational cost of estimating ABCC and ABPC in Section 5.3.
> 2. We have provided more explanation for the hyperparameters of the code in Appendix J.
>
>
>
> ### Concerns and Questions:
>
> **1. ABCC and ABPC cannot be computed directly. The paper may discuss the computational cost of the estimation of ABCC and ABPC.** \
> We thank you for the insightful comment. The computation of the proposed ABPC and ABCC metrics, defined in Equations (3) and (4), are based on the estimated probability density function (PDF) $\tilde{f}_{i}(x)$ and cumulative distribution function (CDF) $\tilde{F}\_{i}(x)$, and numerical integration over $[0,1]$ with $M$ probing points.
> 1. For ABPC, the estimation of PDF is $\tilde{f}\_{i}(x)=\frac{1}{|\mathcal{S}\_i|h}\sum\_{n\in\mathcal{S}\_i}K(\frac{x-\tilde{y}\_{n}}{h})$. The computation complexity of $\tilde{f}_{i}(x)$ for $M$ probing points is $O(MN)$, where $N$ is the number of samples, and the computation complexity of integration is $O(M)$. Therefore, the computation complexity of ABPC is $O(MN)$.
> 2. For ABCC, the computation complexity for CDF estimation with $M$ probing points and integration is $O(MN)$ and $O(M)$, respectively. Thus, the computation complexity of ABCC is also $O(MN)$.

---

> ### Author Response · Authors · 2023-03-30
> **Response to Reviewer tEuV [2/2]**
>
> **2. According to lemma 5.2, the estimation of ABPC requires more samples than ABCC to guarantee the estimation error. But in the experiments, ABPC use 5k samples and ABCC use 10k samples. Why ABCC needs more samples for estimation in the experiments?** \
> We apologize for the confusion. The numbers 5,000 and 10,000 mentioned for ABPC and ABCC are not the number of data samples. Instead, they refer to the number of points used in the integration process for both metrics. Both ABPC and ABCC are based on estimated functions with a range of $[0,1]$. The "sample_n" parameter represents the number of points within the range $[0, 1]$ used for integration. Thus we can set the "sample_n" to a relatively large number for accurate measurement. The default values of 5,000 and 10,000 are not strict requirements. In our experiments, we found that a "sample_n" larger than 100 should be sufficient for accurate results.
>
> We thank you for bringing this to us. We have added the explanation of the parameters in the code and more discussion in Appendix J.
>
>
> We sincerely thank you once again for your time. We hope our response has addressed your concerns, and we are pleased to answer any further questions you may have.
>
> Thanks,\
> Authors

---

### Author Response · Authors · 2023-03-30
**Common Response to Reviwers**

Dear Reviewers,

We sincerely thank you for your time and effort in reviewing our work. In response to your reviews, we have revised and uploaded an updated version to address your concerns. The revisions addressing the concerns of Reviewer ${\color{purple} n71L}$, Reviewer ${\color{orange} YqvS}$ and Reviewer ${\color{blue} tEuV}$ have been highlighted in different colors. We also proofread the paper and fix the grammar error highlighted in red. Additionally, we have provided separate replies to your individual reviews.

Once again, we thank you for your valuable time and effort in reviewing our paper, and we hope that our revisions and direct responses will address your concerns about our work.


Thanks,\
Authors

---

### Decision · Action_Editors · 2023-05-17

**Recommendation:** Accept with minor revision

**Comment:**

The paper highlights shortcomings in existing widely used metrics and also proposes some new ones, derived from desirable criteria for such metrics. It also develops some theory around these and provides some empirical results.

**Audience:**

The paper will be of interest to the sub-community working on fairness in ML. This is a growing community, and it will also have some broader interest.

**Claims And Evidence:**

The paper argues that demographic parity ($\Delta$DP), a widely used metric, for group fariness suffers from shortcomings. The paper then proposes two additional metrics relating to the areas between pdfs and cdfs. While similar metrics have been proposed earlier, the authors do this in a systematic way starting from desired criteria. The paper also provides some empirical results on standard publicly-available datasets.